# Anatomy of nerve fiber bundles at micrometer-resolution in the vervet monkey visual system

Hiromasa Takemura[1,2†*], Nicola Palomero-Gallagher[3,4,5†*], Markus Axer[3], David Gräßel[3], Matthew J Jorgensen[6], Roger Woods[7], Karl Zilles[3,8†]

[1]Center for Information and Neural Networks (CiNet), National Institute of Information and Communications Technology, and Osaka University, Osaka, Japan; [2]Graduate School of Frontier Biosciences, Osaka University, Osaka, Japan; [3]Institute of Neuroscience and Medicine INM-1, Research Centre Jülich, Jülich, Germany; [4]Department of Psychiatry, Psychotherapy and Psychosomatics, Medical Faculty, RWTH Aachen, Aachen, Germany; [5]C. & O. Vogt Institute for Brain Research, Heinrich-Heine-University, Düsseldorf, Germany; [6]Department of Pathology, Section on Comparative Medicine, Wake Forest School of Medicine, Winston-Salem, United States; [7]Ahmanson-Lovelace Brain Mapping Center, Departments of Neurology and of Psychiatry and Biobehavioral Sciences, David Geffen School of Medicine, UCLA, Los Angeles, United States; [8]JARA - Translational Brain Medicine, Aachen, Germany

*For correspondence:
htakemur@nict.go.jp (HT);
n.palomero-gallagher@fz-juelich.de (NP-G)

†These authors contributed equally to this work

Competing interests: The authors declare that no competing interests exist.

**Abstract** Although the primate visual system has been extensively studied, detailed spatial organization of white matter fiber tracts carrying visual information between areas has not been fully established. This is mainly due to the large gap between tracer studies and diffusion-weighted MRI studies, which focus on specific axonal connections and macroscale organization of fiber tracts, respectively. Here we used 3D polarization light imaging (3D-PLI), which enables direct visualization of fiber tracts at micrometer resolution, to identify and visualize fiber tracts of the visual system, such as stratum sagittale, inferior longitudinal fascicle, vertical occipital fascicle, tapetum and dorsal occipital bundle in vervet monkey brains. Moreover, 3D-PLI data provide detailed information on cortical projections of these tracts, distinction between neighboring tracts, and novel short-range pathways. This work provides essential information for interpretation of functional and diffusion-weighted MRI data, as well as revision of wiring diagrams based upon observations in the vervet visual system.

## Introduction

Over the last decades, the architecture and function of the cortical areas in the primate visual system has been extensively studied. A number of studies have proposed theories on the organization of these areas and visual processing streams. For example, one key theory categorized visual areas into dorsal and ventral streams, which are involved in the control of actions and in the identification of objects, respectively (*Goodale and Milner, 1992*; *Ungerleider and Mishkin, 1982*). However, increasing evidence supports the notion that these two streams are not strictly independent, and interact with each other at different levels (macaque study, *Tolias et al., 2005*; human studies, *Grill-Spector et al., 1998*; *Konen and Kastner, 2008*; *Freud et al., 2016*; *Milner, 2017*). Furthermore, the concept of an intermediate visual stream has also been proposed (macaque study, *Boussaoud et al., 1990*; human study, *Weiner and Grill-Spector, 2013*; see *Binkofski and Buxbaum, 2013*; *Rizzolatti and Matelli, 2003* for reviews including works in both macaques and

humans). In contrast, the anatomical organization of single fiber tracts in the white matter of the visual system was less frequently studied (*Rockland, 2013*; *Rockland, 2018*; *Yeatman et al., 2014*; *Caspers et al., 2015*; *Takemura et al., 2019b*), although this knowledge is crucial to understand the structural organization of the network subserving the visual processing streams or interactions between streams.

The white matter tracts of the visual system have been studied using several different methods. Classical neuroanatomists have analyzed approximate shape, position and trajectories of human white matter tracts using dissection methods (*Sachs, 1892*; *Déjerine, 1895*). Although the trajectories of early visual pathways (optic nerve, optic tract and optic radiation) have been established, the existence and organization of association fibers in the occipital lobe have been controversial among classical neuroanatomists (*Yeatman et al., 2014*). Later, axonal tract tracing has been widely used to investigate anatomical connections in the visual system of non-human primates (*Lanciego and Wouterlood, 2011*; *Lanciego and Wouterlood, 2020*; *Kennedy et al., 2013*; *Rockland, 2020*). A comprehensive overview of axonal tracing on major white matter tracts in macaque monkey was provided by *Schmahmann and Pandya, 2006*. Based on axonal tracing, wiring diagrams on visual cortico-cortical connectivity have been proposed (*Boussaoud et al., 1990*; *Felleman and Van Essen, 1991*; *Wallisch and Movshon, 2008*). Over the last decade, there has been a resurgent interest in studying properties of white matter tracts in the human visual system using diffusion-weighted MRI (dMRI) and tractography, which can demonstrate position and trajectories of large and expected connections in living brains (*Mori et al., 1999*; *Conturo et al., 1999*; *Catani et al., 2002*; *Behrens et al., 2003*; *Wakana et al., 2004*; *Sherbondy et al., 2008*; *Wandell, 2016*; *Rokem et al., 2017*). The advancement of dMRI acquisition and analysis methods led developments on the atlases of human major fiber tracts (*Mori et al., 2008*; *Catani and Thiebaut de Schotten, 2008*; *Catani and Thiebaut de Schotten, 2012*; *Yeh et al., 2018*) and automated procedures to analyze those tracts based on dMRI data (*Zhang et al., 2008*; *Yendiki et al., 2011*; *Yeatman et al., 2012*; *Yeatman et al., 2018*; *Wassermann et al., 2016*; *Wasserthal et al., 2018*; *Warrington et al., 2020*).

Despite the collections of dissection, tracer and dMRI studies on the visual system, we do not fully understand detailed spatial organization of the white matter tracts in the visual system because there remains a large gap between studies performed by different methods (*Takemura et al., 2019b*; *Rushmore et al., 2020*). Specifically, while tracers are well suited to measure specific connections from or to injection sites, this method is not able to visualize the entire fiber tracts. On the other hand, while dMRI is well suited for measuring approximate position and trajectories of major fiber tracts, it does not have enough resolution to precisely measure termination of fiber tracts in cortical gray matter (*Reveley et al., 2015*). Therefore, there is a large gap between findings on cortico-cortical connectivity from tracer studies and findings on white matter tracts from dMRI studies. Moreover, there are many remaining questions regarding spatial organization of white matter tracts, since it is difficult to precisely measure such organization using any of the aforementioned methods. For example, it is not yet clear how much the vertical occipital fascicle (VOF; *Yeatman et al., 2014*) is an independent fascicle from the inferior longitudinal fascicle (ILF) in the macaque (*Schmahmann and Pandya, 2006*; *Takemura et al., 2017*). Moreover, the spatial organization of neighboring tracts, such as the stratum sagittale (SS) and the ILF, has been controversially discussed among investigators (*Schmahmann and Pandya, 2006*). We also note that not all studies reported the same fiber tracts or proposed identical definitions of fiber tracts (*Schmahmann and Pandya, 2006*; *Yeatman et al., 2014*). These ambiguities in the literature result partly from different methods used for each study (dissection, dMRI and tracer) because these methods have their own advantages and limitations. A study visualizing whole fiber bundles with higher spatial resolution seems necessary to fill a gap between different methods and to establish our understanding on the detailed spatial organization of visual white matter tracts.

Therefore, we used 3D polarized light imaging (3D-PLI) in post-mortem brains of vervet monkeys. This method provides an overview on the orientation and localization of all fiber tracts and their components in the white matter (*Axer et al., 2011a*; *Zeineh et al., 2017*; *Zilles et al., 2016*; *Caspers and Axer, 2019*). Moreover, 3D-PLI provides visualization of fiber tracts simultaneously at microscopic resolution (in-plane resolution: 1.3 μm) and in the entire undissected brain, and thus opens an avenue to fill the gap between dMRI and tracer studies by demonstrating the anatomical

ground truth of the structures. Moreover, 3D-PLI does not require any fiber staining procedures, but uses the biophysical properties of myelinated fibers, i.e. their birefringence.

We studied several major white matter tracts in the visual system of the vervet monkey, such as SS, ILF, and VOF. Although these tracts have already been identified (macaque studies, *Schmahmann and Pandya, 2006*; *Schmahmann et al., 2007*; *Takemura et al., 2017*; human studies, *Catani et al., 2003*; *Toosy et al., 2004*; *Catani and Thiebaut de Schotten, 2008*; *Catani and Thiebaut de Schotten, 2012*; *Kamali et al., 2014*; *Yeatman et al., 2014*), these studies either did not provide direct evidence of the underlying anatomical structure, or are prone to methodical limitations (see Discussion). We could also investigate further structures of the visual system, i.e. tapetum, stratum calcarinum and dorsal occipital bundle (dOB). While some of these structures have already been described (macaque study, *Schmahmann and Pandya, 2006*; human studies, *Sachs, 1892*; *Déjerine, 1895*; *Forkel et al., 2015*; *Bugain et al., 2020*; study on both macaque and human, *Oishi et al., 2011*), their relations to different cortical areas remained largely unknown.

Here, we present evidence for the anatomy of SS, ILF, VOF, tapetum, stratum calcarinum and dOB as distinct units of white matter architecture in the vervet visual system. We present further evidence for the segregation of VOF from ILF fibers, distinction between SS and ILF.

## Results

*Figure 1* depicts descriptions of major tracts in the primate visual system, the SS, the ILF and the VOF as defined in previous human dissection studies (*Figure 1A–B*), and macaque tracer

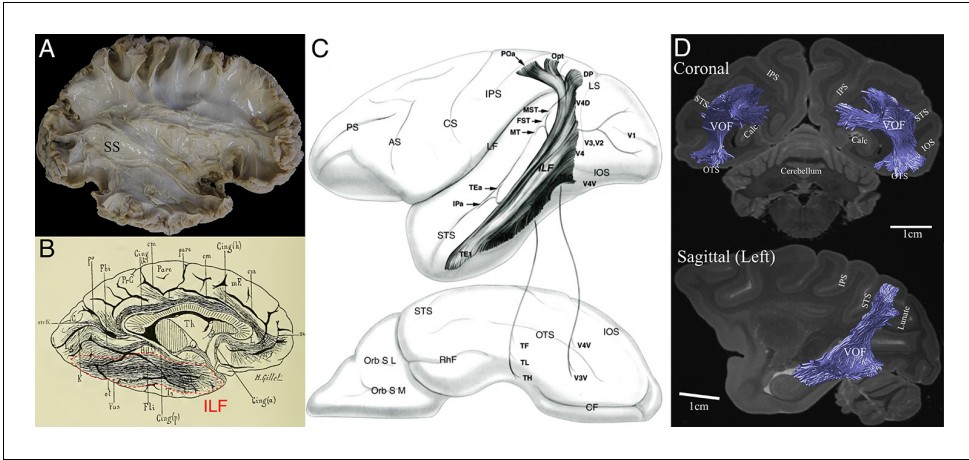

**Figure 1.** Previous studies describing the position and trajectory of major fiber tracts in the primate visual system. (**A**) The position and trajectory of the SS identified by Klingler's dissection method in the human brain (provided by courtesy of Sabine Wittschonnke). The SS is visible as a large fiber bundle located in the medial portion of the occipital white matter and travelling through an anterior-to-posterior axis. It is difficult to identify the precise termination of SS fibers by this method. (**B**) Camera lucida drawing of the human ILF in the classical dissection work by *Déjerine, 1895*. The ILF (highlighted by red dotted lines) was described as a tract connecting occipital and inferotemporal cortex. (**C**) Schematic diagram of the macaque ILF based on tracer experiments (*Schmahmann and Pandya, 2006*). Similar to *Déjerine, 1895*, the ILF was described as a tract connecting occipital and inferotemporal cortex. (**D**) The VOF in the macaque monkey identified by dMRI (*Takemura et al., 2017*). Similar to a definition in classical dissection works (*Yeatman et al., 2014*), the VOF was identified as a tract connecting dorsal and ventral occipital cortex and located lateral to the SS.

(*Figure 1C*) and dMRI (*Figure 1D*) studies. While these studies revealed the approximate position and trajectory of each tract (see *Table 1* for definition of these tracts), none of them revealed the spatial organization of the whole single fiber tract in question at a micrometer resolution. Therefore, we aimed to investigate the detailed spatial organization of these tracts, and other tracts not well described in *Figure 1*, using 3D-PLI approach on the vervet monkey visual system.

## Stratum Sagittale (SS)

The SS is clearly visible in sagittal sections as a fiber tract running between the lateral geniculate nucleus (LGN) and the primary visual cortex (V1). Within most medially located levels, the course of the SS follows a rostro-caudal and slightly upwards shifted direction (*Figure 2A,D*). We identified the region where geniculo-cortical fibers leave the LGN and merge with the most rostral portion of the SS (*Figure 2B,D*). The transmittance image of this region (*Figure 2C*) visualizes structures with a high myelin density and/or an orientation perpendicular to the plane of sectioning in black, whereas regions with higher cell body density (such as the caudate nucleus and pulvinar) appear in grey. In addition to the geniculo-cortical fibers, we also found fibers from the lateral pulvinar which turn into the SS (*Figure 2B*). Thus, it is very likely that fibers from the LGN and pulvinar merge into the SS. At a more lateral level SS is also clearly visible and reaches V1 (*Figure 3A*). Only 0.2 mm more lateral (*Figure 3C*), the VOF crosses the SS nearly vertically. The SS and the VOF are sharply distinguishable by virtue of the different fiber orientations. Even more laterally, a temporal and occipital part of the SS appear (*Figure 4A,B*), which disappear at the most lateral level (*Figure 4C*).

The rostro-caudal course of the SS can also be followed in coronal sections. *Figures 5–10* show a series of coronal sections from rostral to caudal levels. SS appears as a black structure in the fiber orientation map (FOM) images, because the orientation of its fibers is perpendicular to the plane of sectioning, and thus, rostro-caudally directed. At rostral levels, SS is sandwiched between the short-range fibers along the superior temporal sulcus laterally and the ventral subcortical bundle (vSB) medially (*Figure 5D–E*). At this level, the geniculo-cortical fibers cross the vSB and reach the SS laterally by diving into a sagittal orientation (*Figure 5B–C*). When moving more caudally, the vSB is replaced by fibers of the striatal bundle (StB; *Figure 6*). Again more caudally, the SS is delimited medially by the tapetum (T) and laterally by the VOF (*Figures 7A* and *8A*). Even more occipitally, the VOF is replaced dorsally by transverse occipital fibers and ventro-laterally by a fiber bundle running between V1 and V3v (*Figure 9A*). This fiber bundle underlies the lateral portion of V1, and

**Table 1.** Summary of observations of major visual white matter tracts in 3D-PLI data of the vervet monkey brain.

| Tract | Definition and trajectory | Origins/terminations visible in 3D-PLI data (figure number) |
|---|---|---|
| SS | Parasagittally oriented white matter tract in posterior part of brain (*Sachs, 1892*). Thought to include thalamo-cortical fibers (*Schmahmann and Pandya, 2006*). | Includes fibers between LGN and V1 (*Figures 2*, *3* and *10*). Also includes other fibers with termination/origin in lateral pulvinar (*Figure 2*) and V2v (*Figure 10*). Other fibers turning into the SS also described previously (*Schmahmann and Pandya, 2006*). |
| ILF | White matter tract travelling along lateral gyri in the temporal lobe (*Burdach, 1822*). Composes major associative connection between occipital and temporal lobe (*Catani et al., 2003*). | Can be divided into fibers dorsal to the SS and those ventral to the SS (*Figures 3–4*). At anterior end, both dorsal and ventral fibers terminate at TE (*Figures 3–4*). Posterior termination of the ventral ILF appears at V3v, V4v and TEO (*Figure 4*). At posterior end, ventral fibers merged with VOF fibers (*Figure 4*). |
| VOF | Association white matter tract located laterally adjacent to the SS and connecting dorsal and ventral occipital cortex (*Yeatman et al., 2014*). | Dorsal terminations found at V3d, V3A, V4d, V4d and DP (*Figures 3–4* and *7–8*). Ventral terminations found at V3v, V4v and TEO (*Figure 7* and *Figure 7—figure supplement 1*). Some fibers merged with ventral ILF fibers near ventral extrastriate cortex (*Figure 4*). |
| dOB | Association white matter tract located between the SS and annectant gyrus (*Schmahmann and Pandya, 2006*). | Posterior termination in V3A (*Figure 9*). Connects medial and lateral parts of cortex in the intraparietal sulcus. Difficult to precisely identify all dOB terminations since its merges with VOF and U-fibers (*Figure 8*). |
| tapetum | Slender caudal and lateral component of the corpus callosum (*Burdach, 1822*; *Clarke and Miklossy, 1990*; *Schmahmann and Pandya, 2007*). | Anterior origin is splenium corporis callosi (*Figure 11*). Posterior terminations were not always identifiable, but may terminate in the border tuft region or area prostriata (*Figures 8–9* and *11*). |
| stratum calcarinum | U-fiber layer of the calcarine sulcus (*Déjerine, 1895*). | Connects dorsal and ventral lips of calcarine sulcus, which correspond to border tuft region and area prostriata (*Figures 7–11*). |

curves around the fundus of the inferior occipital sulcus (IOS), where it is located between U-fiber bundles under areas V2v and V3v and the SS. This fiber bundle can be followed to the ventromedial side of the hemisphere, where area V3v could be identified between the collateral (COS) and occipito-temporal (OTS) sulci. At the most occipital level, the fibers within SS change their orientation from the previously observed strictly sagittal orientation and bend into a medio-lateral direction (*Figure 10A*). Because of the intermingling of the SS fibers and those of the above mentioned fiber bundle under V1 and V2v, the colour of the crossing fibers becomes dark (*Figure 10C*). Nonetheless, the fibers of the fiber bundle under V1 and V2v can still be differentiated. Thus, the ventro-lateral border of SS begins to dissolve, while its fibers reach out to V1 and V2v. Contrastingly, the dorso-lateral border of SS is still clearly visible and delimited by the stratum calcarinum medially and the fiber bundle under V1 laterally (*Figure 10B*). On the medial side, the ventral portion of SS is delimited by the tapetum (*Figure 10C*).

## Inferior Longitudinal Fascicle (ILF)

A series of sagittal sections (*Figures 3* and *4*) demonstrates the ILF, a fiber bundle extending between the preoccipital gyrus and the infero-temporal cortex. The ILF appears to be split into a dorsal and a ventral part with the SS in between (*Figures 3A*, *4A–C and E–F*). Thus, 3D-PLI data provide evidence for a hitherto unknown complexity of the white matter in the inferior temporal gyrus. The SS is bordered dorsally and ventrally by two separate portions of the ILF. This segregation between ILF and SS is based on the slightly different orientations of ILF and SS fibers (*Figure 4E–F*). ILF and SS are separated by very thin black septa. While the ILF exists as an independent fascicle in the inferior temporal gyrus, near the fundus of the OTS the ventral portion of the ILF merges into the VOF (*Figure 4A–B,D*). The ventral portion of the ILF turns around the OTS and seems to terminate in V3v, V4v and more laterally in TEO (*Figure 4A–C*). This relationship between fiber tracts revealed by 3D-PLI disentangles debates on the potential independence of ILF from SS (see Discussion).

In coronal sections the ILF is situated laterally to the VOF (*Figure 7A*). Above the OTS, ILF may also be visible between V3v and VOF (*Figure 7C*). The most occipital part of ILF is seen in *Figure 8A* and seems to terminate in V3v. At more caudal occipital planes of sectioning ILF is no longer visible. The white matter between inferior occipital sulcus and OTS is largely filled by U-fibers between V2v and V3v or V3v and V4v, respectively (*Figure 9A*).

## Vertical Occipital Fascicle (VOF)

The VOF connects dorsal and ventral occipital cortical areas (*Yeatman et al., 2014*; *Takemura et al., 2016*; *Takemura et al., 2017*; *Takemura et al., 2019b*). At the most medial sagittal section, VOF is clearly separable from SS by its slightly oblique medial to lateral fiber orientation, whereas SS has a stricter rostro-caudal direction (*Figure 3A,C*). It is more difficult to disentangle ILF and VOF fibers (*Figure 4A–D*; *Figure 4—figure supplement 1*). The dorsal part of ILF takes an oblique latero-medial direction, whereas the VOF fibers keep their dorso-ventral direction. The fiber directions of both fascicles approach each other at the level of the superior temporal area in the floor of the superior temporal sulcus (FST; *Figure 4A*). In more lateral sagittal sections, the superior portion of the ILF stops below the temporal cortex (*Figure 4B–C*), and is completely separated from VOF by a dark area, at the rostral border between the temporo-occipital region TEO and other temporal areas. This black region indicates the bending of the ILF fibers into a plane vertical to the sagittal one. VOF reaches most dorsally the white matter of the preoccipital gyrus, where areas V4d and DP are found (*Figures 3A–B* and *4A–C,G*). On coronal sections VOF is visible immediately lateral to SS and clearly posterior to the splenium corporis callosi (see *Figure 7—figure supplement 1*). In *Figure 7*, the VOF appears to be divided into two components: one is lateral and the other one is medial to superior temporal sulcus. This division of the VOF is also maintained at a more caudal level (*Figure 8*). The lateral component extends between extrastriate areas (V3d, V3A, V4d, and V4t) below the lunate sulcus. In contrast, the medial component runs lateral to the SS and reaches dorsal extrastriate areas (area DP). The terminal branches of VOF take different orientations caused by the orientations of the respective gyri. At most caudal levels (*Figures 9–10*) the VOF is no longer visible.

In contrast to the relationship of VOF to dorsal visual areas, that of the ventral part of VOF is more challenging, because U-fibers around the OTS mask the termination of VOF fibers before they

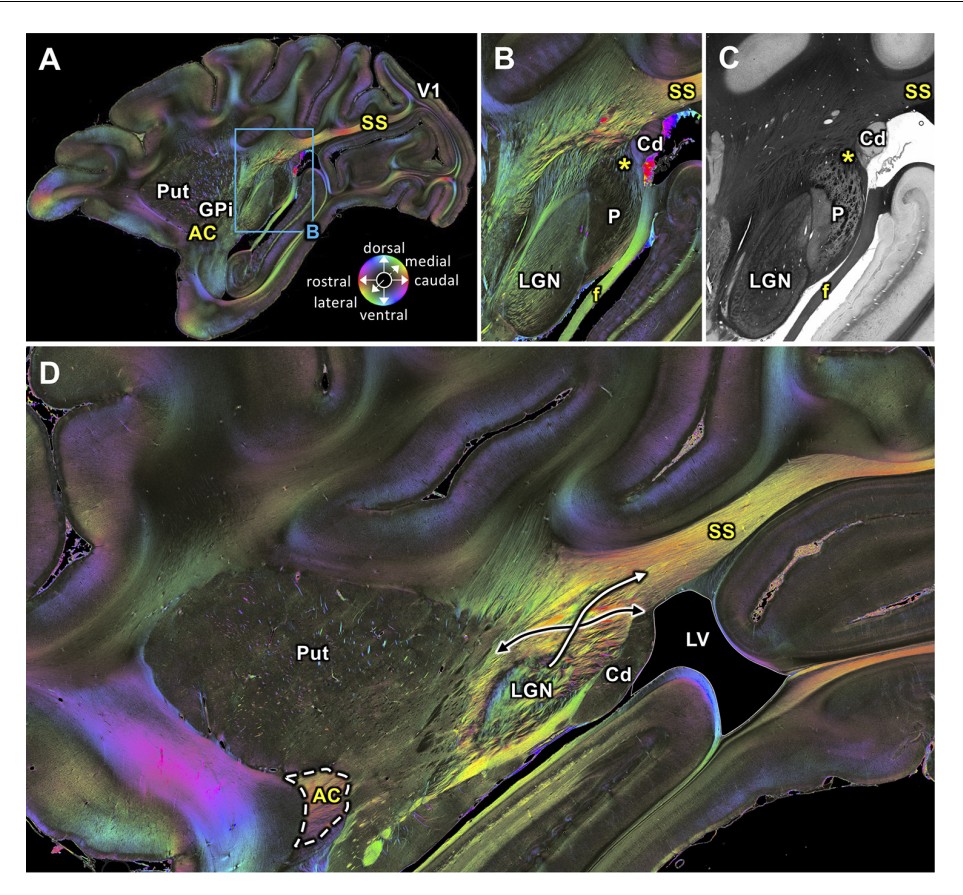

**Figure 2.** Medial sagittal sections through the left hemisphere of the vervet monkey brain (ID1947; **A-C** section #249, which is medial to **D** section #210) including the LGN. (**A, B, D**) FOMs; (**C**) transmittance image. (**A**) Overview with fiber tract. (**B-C**) Detailed magnifications of the rectangle in **A**. The sphere (in **A**) indicates the color coding of 3D fiber orientation in each pixel. (**D**) Zone in which SS-fibers (white arrow) leave the LGN and cross with fibers running between the caudate nucleus and the putamen (black arrow). AC: anterior commissure, Cd: caudate nucleus, f: fornix/fimbria hippocampi, GPi: globus pallidus pars interna, LGN: lateral geniculate nucleus, LV: lateral ventricle, P: pulvinar, Put: putamen, SS: stratum sagittale, V1: primary visual cortex. Asterisks indicate SS fibers leaving the pulvinar.

reach into the cortical gray matter (*Figures 7A,C* and *8A*). This observation is consistent with a previous study showing that an intense signal from superficial U-fiber systems constitutes a challenge for identification of the cortical termination of the longitudinal fasciculus (*Reveley et al., 2015*). However, it is likely that VOF may terminate at cortical regions around the OTS, such as V3v and V4v (*Figure 7A*) or TEO (*Figure 7—figure supplement 1*).

### Tapetum and stratum calcarinum

In a series of coronal images (*Figures 7A*, *8A*, *9A,C* and *10C*), the tapetum is detectable. The stratum calcarinum, a U-fiber system underlying the intracalcarine portion of V1, can be seen on the same images. These fiber tracts are located immediately adjacent and medial to the SS. Importantly, 3D-PLI data reveal that the stratum calcarinum is distinct from the tapetum. Specifically, the tapetum and stratum calcarinum form distinct outer and inner layers of fibers beneath the primary visual cortex in the calcarine sulcus (*Figure 9A,C*). Furthermore, the FOM images also suggest that while the tapetum fibers run lateral and medial to the ventricle and ependymal (*Figure 9—figure supplement 1*), the stratum calcarinum runs medial to the lateral ventricle.

In the coronal section shown in *Figure 9*, the dorsal part of the tapetum is visible dorsally of the dorsal part of the stratum calcarinum and ventrally of the dorsal part of the SS (*Figure 9C*). The SS,

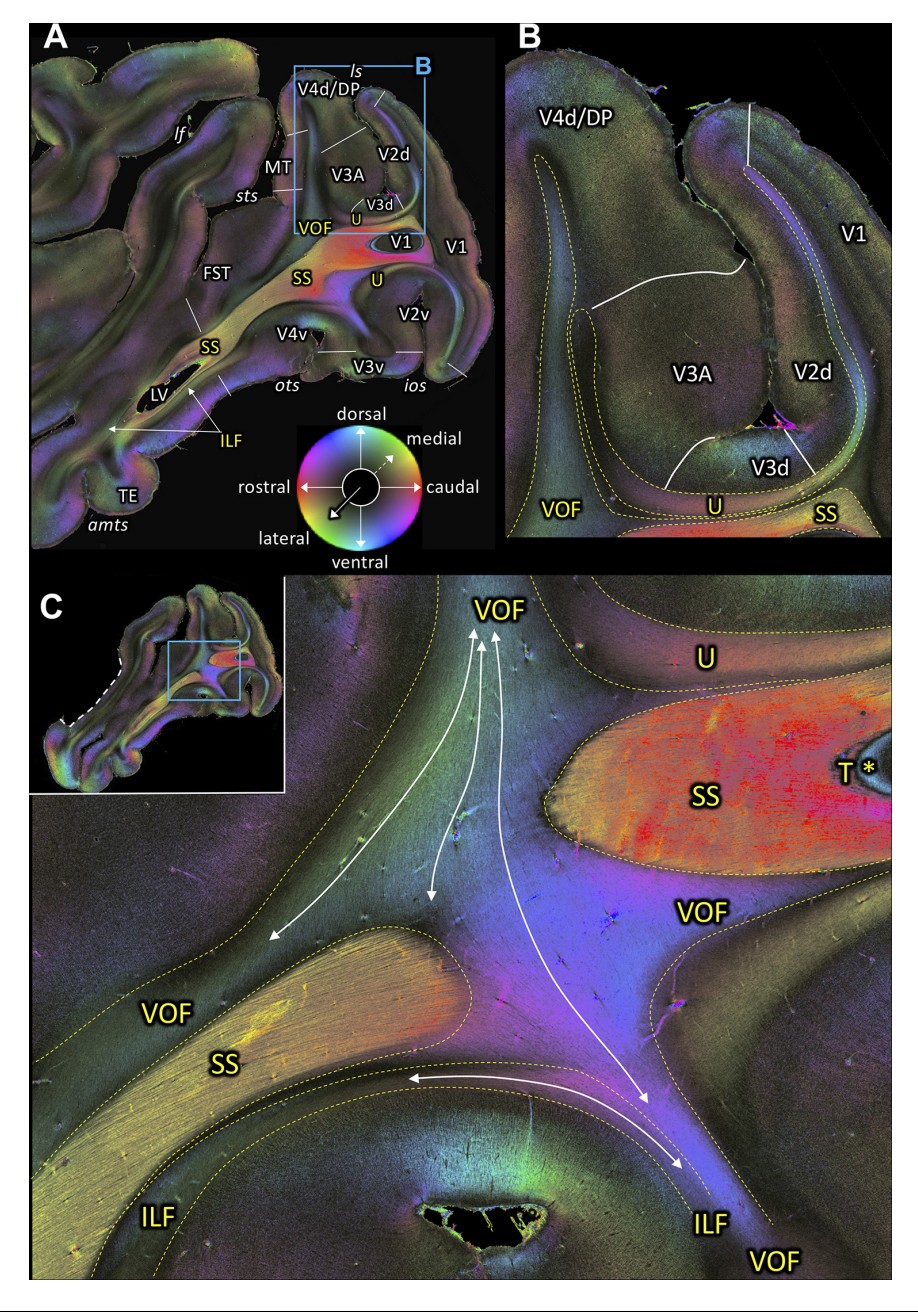

**Figure 3.** Fiber orientation maps of sagittal sections through the left hemisphere of brain ID1947 (**A** section #160 lies medial to **C** section #157). This series is lateral to that of *Figure 2*. The sphere indicates the color coding of 3D fiber orientation in each pixel. (**A**) Overview with fiber tract and cortical areas. (**B**) A magnification of the rectangle in **A**. (**C**) Fiber orientation map of the region where the VOF curves around the SS during its course between the preoccipital and inferior temporal gyri. The small whole-section image on the top left depicts magnified region as a rectangle and the site where the superior temporal gyrus was separated from the rest of the brain as a white dashed line. amts: anterior midtemporal sulcus, DP: dorsal prelunate area, FST: superior temporal area in the floor of the superior temporal sulcus, ILF: inferior longitudinal fascicle, ios: inferior occipital sulcus, lf: lateral fissure, ls: lunate sulcus, LV: lateral ventricle, MT: middle temporal area, ots: occipito-temporal sulcus, SS: stratum sagittale, sts: superior temporal sulcus, T: tapetum, TE: area TE, U: U-fibers, V1: primary visual cortex, V2d: secondary visual cortex dorsal part, V2v: secondary visual cortex ventral part, V3A: visual area V3A, V3d: visual area three dorsal part, V3v: visual area three ventral part, V4d: visual area four dorsal part, V4v: visual area four ventral part, VOF: vertical occipital fascicle. Asterisk indicates the stratum calcarinum.

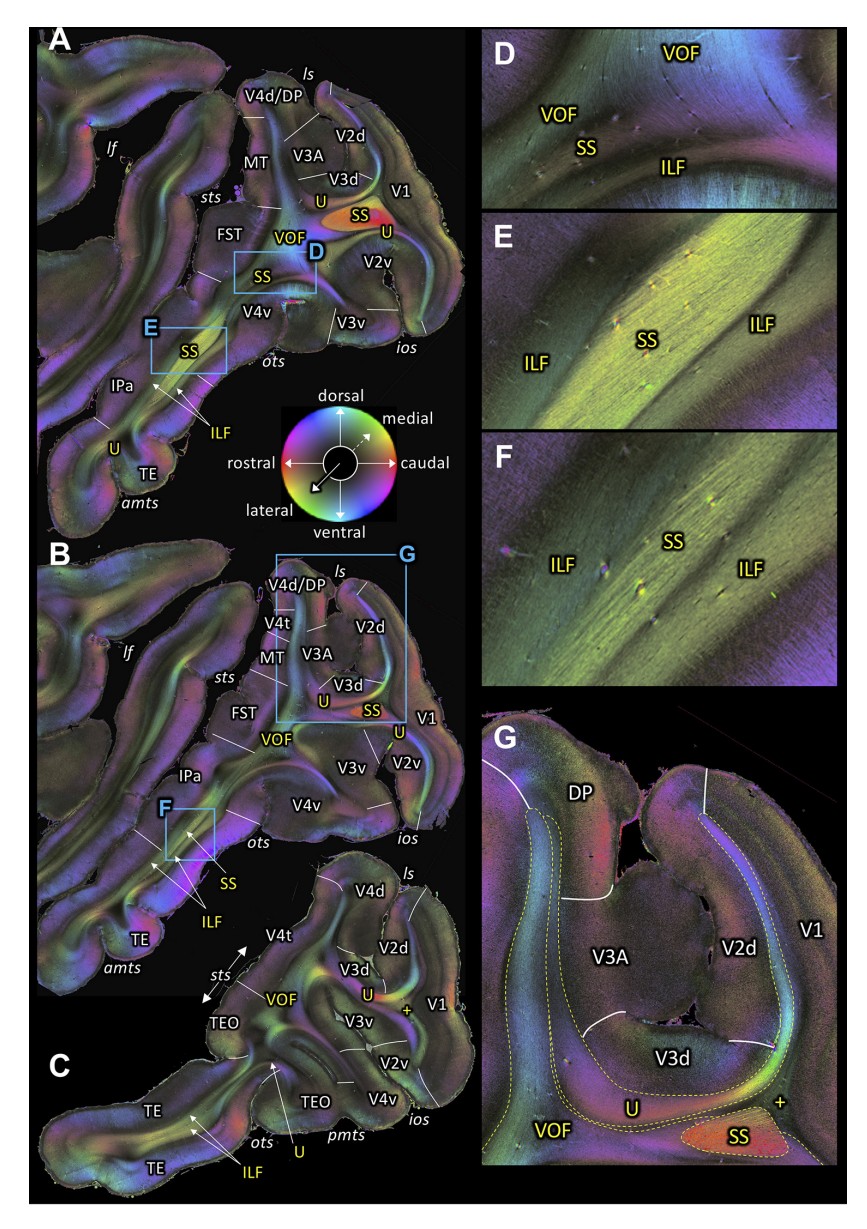

**Figure 4.** Fiber orientation maps of sagittal sections through the left hemisphere of brain ID1947 from medial to lateral (**A-C**). (**A-C**) Overview with fiber tract and cortical areas. (**A**) Section #151, (**B**) section #143, (**C**) section #97. This series is lateral to that of *Figure 3*. (**D-G**) Magnifications of the respective rectangles in A and B. The sphere indicates the color coding of 3D fiber orientation in each pixel. amts: anterior midtemporal sulcus, DP: dorsal prelunate area, FST: superior temporal area in the floor of the superior temporal sulcus, ILF: inferior longitudinal fascicle, ios: inferior occipital sulcus, IPa: area IPa in the fundus of the superior temporal sulcus, lf: lateral fissure, ls: lunate sulcus, MT: middle temporal area, ots: occipito-temporal sulcus, pmts: posterior middle temporal sulcus, SS: stratum sagittale, sts: superior temporal sulcus, TE: area TE, TEO: area TEO, U: U-fibers, V1: primary visual cortex, V2d: secondary visual cortex dorsal part, V2v: secondary visual cortex ventral part, V3A: visual area V3A, V3d: visual area three dorsal part, V3v: visual area three ventral part, V4d: visual area four dorsal part, V4t: visual area four transitional area, V4v: visual area four ventral part, VOF: vertical occipital fascicle. +: fibers underlying lateral V1 (stratum extra-calcarinum).

The online version of this article includes the following figure supplement(s) for figure 4:

**Figure supplement 1.** Magnification of a sagittal section (#151 of brain ID1947) showing the intricate spatial relations of VOF.

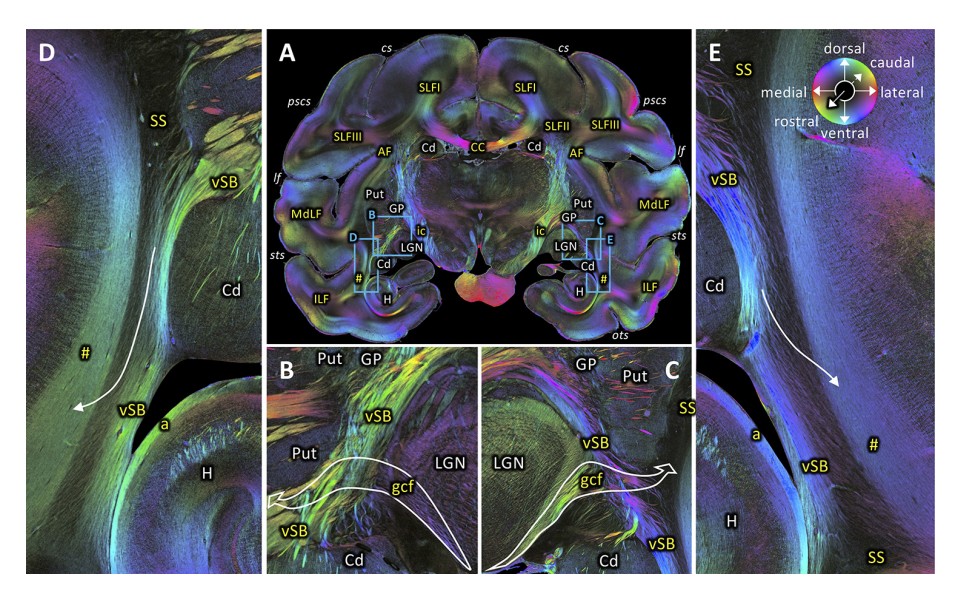

**Figure 5.** Rostral beginning of the SS. (A) Coronal section at the level of the lateral geniculate body (section #582, both left and right hemispheres, brain ID1818). (B-C) Magnifications of rectangles B (right hemisphere) and C (left hemisphere) in A. Geniculo-cortical fibers (gcf; course highlighted by arrow) leave the lateral geniculate body, cross over the ventral subcortical bundle (vSB), and merge with the SS. (D-E) Magnifications of rectangles D (right hemisphere) and E (left hemisphere) in A. Lower part of the SS, bordered medially by the vSB, and laterally by the fibers along superior temporal sulcus (marked as hashtag). Fibers between the vSB and the fibers along superior temporal sulcus cross the SS (arrow). The sphere indicates the color coding of 3D fiber orientation in each pixel. a: alveus, AF: arcuate fascicle, CC: corpus callosum, Cd: caudate nucleus, cs: central sulcus, gcf: geniculo-cortical fibers, GP: globus pallidus, H: hippocampus, ic: internal capsule, ILF: inferior longitudinal fascicle, lf: lateral fissure, LGN: lateral geniculate nucleus, MdLF: middle longitudinal fascicle, ots: occipito-temporal sulcus, pscs: posterior subcentral sulcus, Put: putamen, SLFI-III: superior longitudinal fascicle (parts I to III), SS: stratum sagittale, sts: superior temporal sulcus, vSB: ventral subcortical bundle. Hashtag indicates fibers along superior temporal sulcus.

the tapetum and the stratum calcarinum form three distinct layers of fibers surrounding the primary visual cortex (*Figure 9C*). While the tapetum and the stratum calcarinum have very similar fiber orientations, the presence of a dark area between them suggests that they are distinct fiber tracts (*Figure 9C*). At higher magnification (*Figure 9—figure supplement 1, B*), the tapetum is split by the lateral ventricle into a part ventrally and dorsally of the ventricle and the vestigial ependyma. The ventral part of the tapetum underlies the stratum calcarinum and neighbors the SS and again, is split by the ventricle and the vestigial ependyma (*Figure 9—figure supplement 1, C*). However, in the most occipital coronal section (*Figure 10*), the tapetum disappears in the dorsal portion of the white matter surrounding the calcarine sulcus, and the stratum calcarinum is directly interleaved between V1 and SS (*Figure 10B*). In the ventral part of the white matter around V1, the tapetum is still visible as a separate structure between the stratum calcarinum and SS (*Figure 10C*).

Fibers directly underlying the lateral part of V1 outside the calcarine sulcus are also visible in the FOM images, which take a position similar to that of the stratum calcarinum. These fibers are labelled by a plus symbol in *Figures 4C,G* and *10*. We call this fiber bundle stratum extra-calcarinum because it does not comply with the definition of the stratum calcarinum.

While tapetum fibers are considered as a part of callosal fibers, there has been a debate, whether it is contiguous with the callosal fibers or part of the longitudinal fascicle (*Schmahmann and Pandya, 2006*; *Schmahmann and Pandya, 2007*; *Forkel et al., 2015*). In posterior sections, fibers of the tapetum surround the depth of the calcarine sulcus between SS and stratum calcarinum as described above. In anterior slices, tapetum fibers run lateral to the lateral ventricle and merge into corpus callosum fibers (*Figure 11B–F*). Thus, the series of 3D-PLI data provide a direct demonstration of the fact that tapetum fibers continue into splenium fibers.

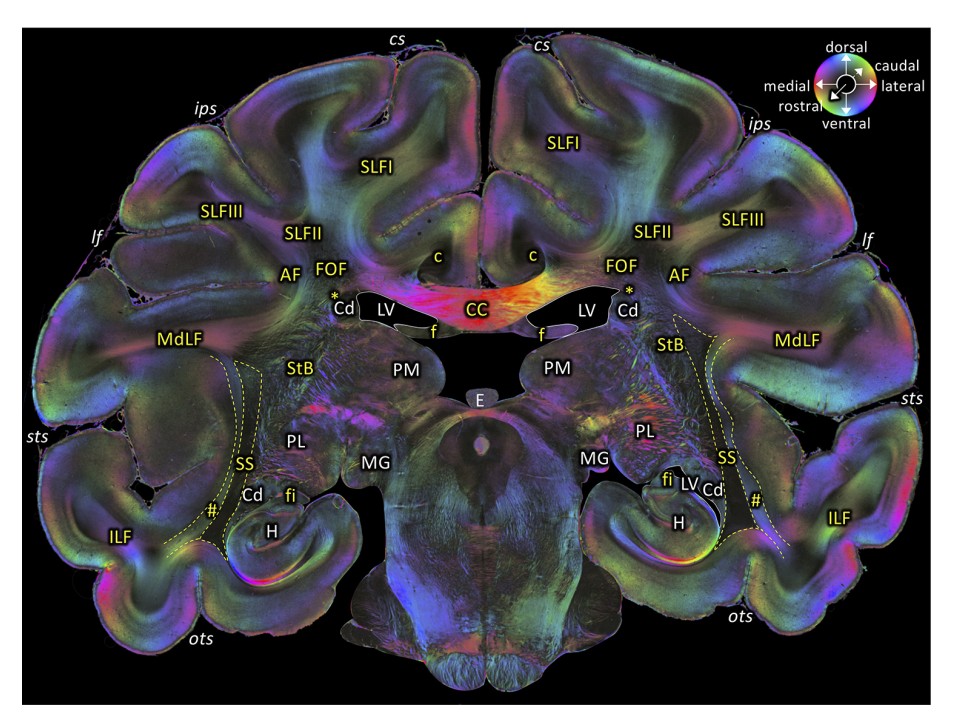

**Figure 6.** Middle portion of the SS as seen in a coronal section (brain ID1818; section #675, both left and right hemispheres, caudal to that shown in *Figure 5*). The sphere indicates the color coding of 3D fiber orientation in each pixel. AF: arcuate fascicle, c: cingulate bundle, CC: corpus callosum, Cd: caudate nucleus, cs: central sulcus, E: epiphysis, f: fornix, fi: fimbria hippocampi, FOF: fronto-occipital fascicle, H: hippocampus, ips: intraparietal sulcus, ILF: inferior longitudinal fascicle, lf: lateral fissure, LV: lateral ventricle, MdLF: middle longitudinal fascicle, MG: medial geniculate body, ots: occipito-temporal sulcus, PL: lateral pulvinar, PM: medial pulvinar, SLFI-III: superior longitudinal fascicle (parts I to III), SS: stratum sagittale, StB: striatal bundle, sts: superior temporal sulcus. Asterisk indicates the Muratoff bundle. Hashtag indicates fibers along superior temporal sulcus.

## Dorsal Occipital Bundle (dOB)

We also could identify the dOB (*Figure 8*). This fiber tract, which was identified by *Schmahmann and Pandya, 2006*, runs in a transverse direction in coronal sections. It connects medial and lateral parts of the cortex in the intraparietal sulcus. The FOM image of a coronal section (*Figure 8A*) shows dOB along the medial wall of the intraparietal sulcus, under annectant gyrus and above the upper segment of SS, consistent with the definition by *Schmahmann and Pandya, 2006*.

## Other transverse fibers

*Figure 9A–B* depicts fibers which are running in a slightly oblique transverse direction between lateral part of V1 and V3A around the medial fundus of the lunate sulcus ('tr' in *Figure 9A–B*). There are other fibers which are running between lateral part of V1 and V3v around the OTS and the COS. At this level, evidence of a vertically running fiber tract (VOF) between dorsal and ventral extrastriate areas cannot be seen.

## U-fibers

In addition to stratum calcarinum, we found a number of U-fibers underlying visual cortical areas. In the ventral extrastriate cortex, we observe U-fibers under the fundi of the inferior occipital, occipito-temporal and collateral which connect neighboring areas (*Figures 7A,C*, *8A*, *9A* and *10C*). It is remarkable that the apex of the U-fibers coincides with borders between cortical areas.

We also observe short-range fibers under the fundus of the superior temporal sulcus (*Figure 7A*; *Figure 7—figure supplement 1*, marked as a hashtag). The fiber orientation of this short-range fiber

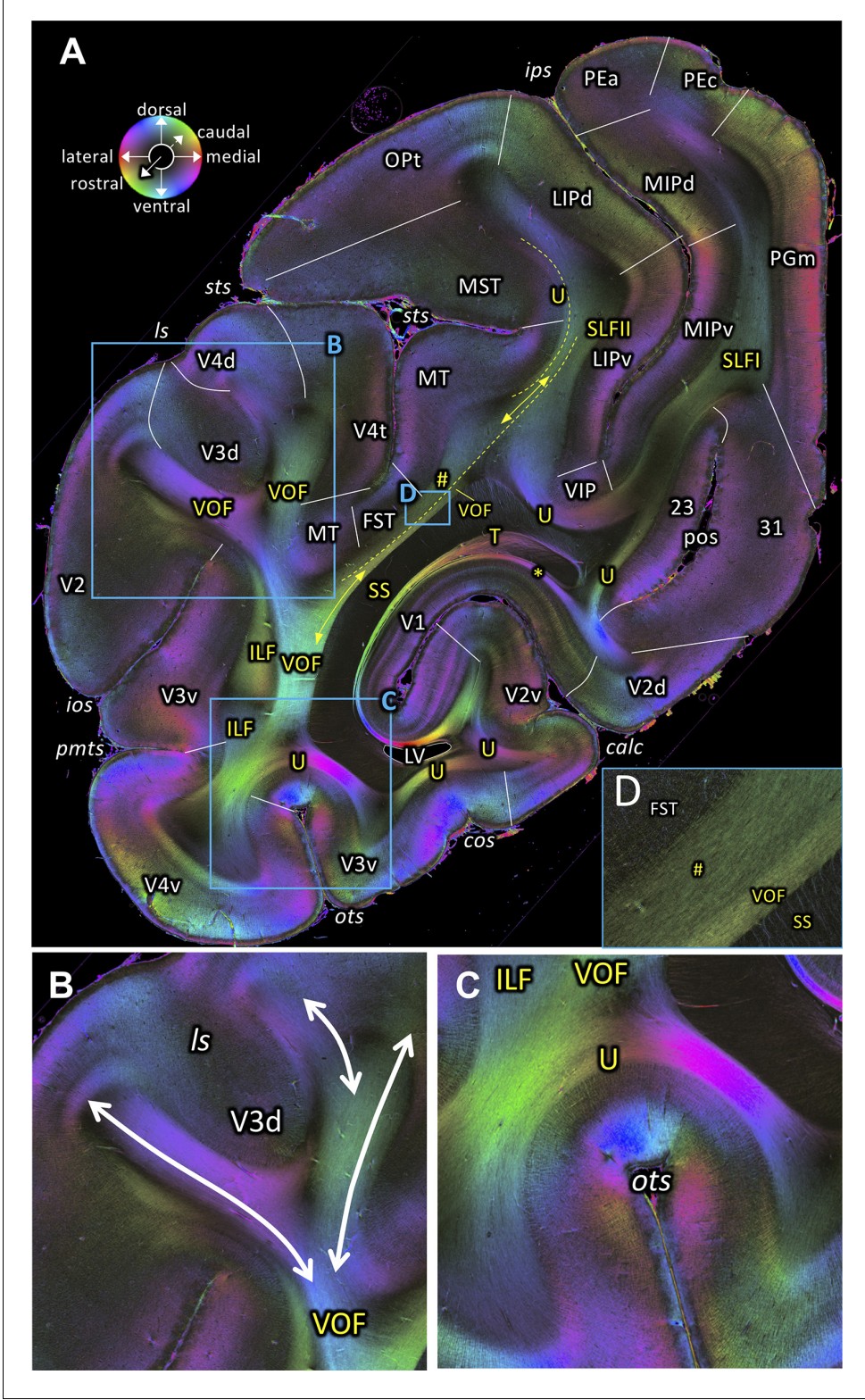

**Figure 7.** Fiber orientation map of a coronal section (brain ID1818; section #860, right hemisphere, caudal to that shown in *Figure 6*). (**A**) Overview with fiber tract and cortical areas. (**B, C and D**) Magnifications of the respective rectangles in **A**. The sphere indicates the color coding of 3D fiber orientation in each pixel. 23: posterior cingulate area 23, 31: posterior cingulate area 31, calc: calcarine sulcus, cos: collateral sulcus, FST: superior temporal area in the floor of the superior temporal sulcus, ILF: inferior longitudinal fascicle, ios: inferior occipital sulcus, ips:

*Figure 7 continued on next page*

*Figure 7 continued*

intraparietal sulcus, LIPd: lateral intraparietal area, dorsal, LIPv: lateral intraparietal area, ventral, ls: lunate sulcus, LV: lateral ventricle, MIPd: medial intraparietal area, dorsal, MIPv: medial intraparietal area, ventral, MST: medial superior temporal area, MT: middle temporal area, OPt: caudal inferior parietal lobule area, , ots: occipito-temporal sulcus, PEa: superior parietal lobule, anterior PE, PEc: superior parietal lobule, caudal PE, PGm: medial parietal area, pmts: posterior middle temporal sulcus, pos: parieto-occipital sulcus, SLFI-II: superior longitudinal fascicle (parts I, II), SS: stratum sagittale, sts: superior temporal sulcus, T: Tapetum, U: U-fibers, V1: primary visual cortex, V2: secondary visual cortex, V2d: secondary visual cortex, dorsal part, V2v: secondary visual cortex, ventral part, V3d: visual area 3, dorsal part, V3v: visual area 3, ventral part, V4d: visual area 4, dorsal part, V4t: visual area 4, transitional area, V4v: visual area 4, ventral part, VIP: ventral intraparietal area, VOF: vertical occipital fascicle. Asterisk indicates stratum calcarinum. Hashtag indicates a short-range fiber along superior temporal sulcus. The online version of this article includes the following figure supplement(s) for figure 7:

**Figure supplement 1.** A fiber orientation map of a coronal section (brain ID1818; section #830, rostral to that shown in *Figure 7*).

**Figure supplement 2.** A fiber bundle along the fundus of the superior temporal sulcus (STS) reported in a previous macaque tracer study.

---

differs only slightly from that of the VOF, but is still clearly visible (*Figure 7D*). In more anterior sections, the distinction between this short-range fiber and the VOF is no longer visible (*Figure 6*).

In posterior section (*Figure 9A–B*), we found relatively large U-fibers under the fundus of the lunate sulcus connecting areas V2d and V3d.

## Discussion

Using an ultra-high resolution 3D-PLI approach, this study demonstrates the existence and organization of projection fibers, callosal fibers, longitudinal association fibers as well as of short association fibers in the vervet visual system. We summarize our observations in *Table 1*, while we note that all cortical terminations of each bundle may not be included in this table, since some were still difficult to identify due to crossing with superficial U-fibers or merging with other tracts. This study provides essential information to clarify the existence, definition and spatial organization of occipital white matter bundles.

### Organization of fiber tracts in the primate visual system revealed by 3D-PLI

#### Stratum Sagittale

While the precise definition of the SS has been debated among investigators, most researchers agree that it includes the optic radiation, which is a projection fiber system connecting the LGN and V1 (*Sachs, 1892*; *Schmahmann and Pandya, 2006*; *Schurr et al., 2018*). Using 3D-PLI, we confirmed the existence of fibers that leave the LGN and merge into the rostral part of the SS (*Figure 2*). Furthermore, we also confirmed that the caudal end of the SS reaches the primary visual cortex (*Figure 3A*). Therefore, 3D-PLI data directly demonstrate and visualize that optic radiation fibers run through the SS.

However, our data also demonstrate an underlying complexity of the SS. For example, we found that fibers from the lateral pulvinar turn into the rostral part of the SS (*Figure 2B*). In the posterior coronal sections, SS fibers may terminate in V2, as well as in V1 (*Figure 10*). Furthermore, coronal FOM data suggest that fibers are not fully perpendicular to the plane in some SS regions. In those areas, we observe some fibers turning into or leaving the SS (e.g. *Figure 9C*). While it is still difficult to fully track the origin of these fibers, their existence suggests that the SS is a system composed by fibers arising from many different parts of cortical areas, while it composes the optic radiation.

The understanding of SS organization is essential for improving functional interpretation of lesion or dMRI studies. For example, the functional role of the connection between pulvinar and extrastriate cortex has been often discussed (*Warner et al., 2012*; *Bridge et al., 2016*; *Baldwin et al., 2017*) and distinguished from those of the optic radiation. However, according to the current observations, it is likely that the pulvinar-extrastriate pathway shares its route with the optic radiation in the SS. While the organization of SS should be further clarified in future works, the present results

already provide important information on how white matter lesions in the SS could affect the optic radiation or pulvinar pathways.

## Inferior Longitudinal Fascicle

The term ILF was initially proposed in a classical human dissection work by *Burdach, 1822* to designate a fiber bundle lying near the lateral ventricle. Burdach described the ILF as being 'arc-like bundles that ascend deep to the lateral gyri of the temporal lobe', which is consistent with the idea that the ILF is a longitudinal association fiber connecting the occipital lobe and inferotemporal cortex. Although in the past the existence of the ILF was questioned (*Tusa and Ungerleider, 1985*), later human dMRI-based tractography (*Catani et al., 2003*) and macaque tracer (*Schmahmann and Pandya, 2006*) studies have demonstrated the existence of the ILF as a longitudinal association fiber. Further analysis of human dMRI together with fMRI or behavioral data suggest a relevance of human ILF with categorical information processing in ventral visual stream (*Gschwind et al., 2012*; *Pyles et al., 2013*; *Scherf et al., 2014*; *Tavor et al., 2014*).

While some of the contentiousness has been largely resolved, questions remain the organization of the ILF in relations to neighboring tracts. For example, while some classical neuroanatomists have often interpreted the ILF as a part of the stratum sagittale externum projection fiber system, other neuroanatomists considered the ILF to be composed solely of association fibers (see *Schmahmann and Pandya, 2006*; *Herbet et al., 2018* for a debate). These different interpretations are largely due to the fact that projection fibers and association fibers are located in close anatomic proximity, so that Klingler's dissection could not distinguish them (*Herbet et al., 2018*). Even to date, detailed organization between SS externum and the ILF has not been revealed yet, since even modern methods such as dMRI-based tractography and tracer-based autoradiography, do not provide the details of similarities and differences of fiber orientation between neighboring tracts at micrometer resolution. In parallel to discussion regarding the SS and the ILF, recent Klingler's dissection and dMRI studies proposed that the ILF has a multi-layered structure, and can be divided into sub-bundles (*Latini, 2015*; *Latini et al., 2017*; *Herbet et al., 2018*; *Panesar et al., 2018*). However, the exact relationship between the SS and possible subcomponents of the ILF remains largely unknown.

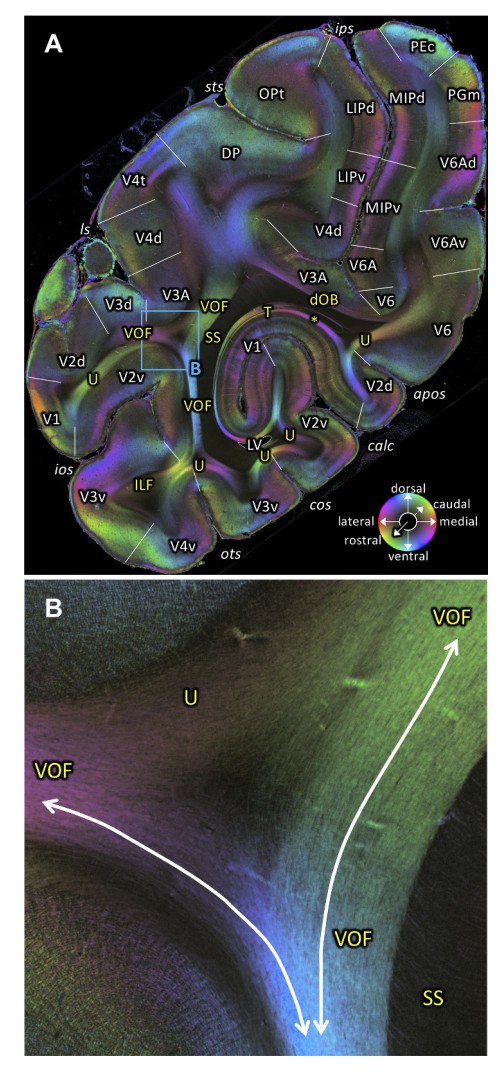

**Figure 8.** Fiber orientation map of a coronal section (brain ID1818; section #898, right hemisphere, caudal to that shown in *Figure 7*). (**A**) Overview with fiber tract and cortical areas. (**B**) Medial and lateral components of the VOF (magnification of the rectangle in **A**). The sphere indicates the color coding of 3D fiber orientation in each pixel. apos: accessory parieto-occipital sulcus, calc: calcarine sulcus, cos: collateral sulcus, dOB: dorsal occipital bundle, DP: dorsal prelunate area, ILF: inferior longitudinal fascicle, ios: inferior occipital sulcus, ips: intraparietal sulcus, LIPd: lateral intraparietal area dorsal part, LIPv: lateral intraparietal area ventral part, ls: lunate sulcus, LV: lateral ventricle, MIPd: medial intraparietal area dorsal part, MIPv: medial intraparietal area ventral part, OPt: caudal inferior parietal lobule area, ots: occipito-temporal sulcus, PEc: superior parietal lobule caudal PE, PGm: medial parietal area, SS: stratum sagittale, sts: superior temporal sulcus, T: Tapetum, U: U-fibers, V1: primary visual cortex, V2d: secondary visual cortex dorsal part, V2v: secondary visual cortex ventral part,

*Figure 8 continued on next page*

*Figure 8 continued*

V3A: visual area V3A, V3d: visual area 3 dorsal part, V3v: visual area 3 ventral part, V4d: visual area 4 dorsal part, V4t: visual area 4 transitional area, V4v: visual area 4 ventral part, V6: visual area 6, V6A: visual area V6A, V6Ad: visual area V6A dorsal part, V6Av: visual area V6A ventral part, VOF: vertical occipital fascicle. Asterisk indicates stratum calcarinum.

The present investigations on 3D-PLI data further confirmed the existence of ILF in the preoccipital gyrus and the infero-temporal cortex (*Figure 3A*; *Figure 4A–F*). Importantly, the ILF can be clearly distinguished from the SS externum in the inferior temporal gyrus because both fiber bundles have distinct inclination angles (*Figure 4E–F*). Thus, while some classical works discuss that the ILF is a part of the external part of the SS, the present investigation suggests that ILF is a distinct fiber bundle from the SS externum. Furthermore, high-resolution 3D-PLI data also revealed that within the temporal lobe the ILF is composed of dorsal and ventral segments, which are separated by the SS (*Figure 4E–F*). Both ILF segments run parallel to the gray matter in temporal cortex (*Figure 4A–F*). While these findings are in line with previous human dissection or dMRI studies proposing a distinction between dorsal and ventral ILF (*Latini, 2015*; *Panesar et al., 2018*), present results provide a direct visualization demonstrating that spatial organization of the ILF with respect to the SS is a key anatomical feature for dividing the ILF into sub-branches. The distinction between dorsal and ventral portions of ILF becomes less obvious at a lateral level where the SS becomes much less visible (*Figure 4C*). These observations in 3D-PLI data will provide useful information for guiding dMRI-based tractography in future studies.

## Vertical Occipital Fascicle

A number of previous visual neuroscience studies proposed that dorsal and ventral extrastriate areas have different roles in visual processing, such that dorsal areas are involved in spatial information processing or guiding action, whereas ventral areas are related to the processing of categorical information (*Ungerleider and Mishkin, 1982*; *Goodale and Milner, 1992*). Since the VOF is a fiber bundle connecting areas of the dorsal and ventral extrastriate cortex, it is a crucial fiber tract to understand how the visual system integrates spatial and categorical information (*Takemura et al., 2016*). While the VOF was described in dissection studies carried out in the late 19th century, its existence has been debated among neuroanatomists and largely ignored in the neuroscience literature (*Yeatman et al., 2014*). It has been sometimes considered that visual information from the primary visual cortex reaches parietal and inferotemporal cortices through largely separated anatomical pathways (*Morel and Bullier, 1990*). Recently, there are resurgent interests concerning the VOF since dMRI-based tractography studies demonstrate it in living human brains (*Yeatman et al., 2013*; *Yeatman et al., 2014*; *Duan et al., 2015*; *Takemura et al., 2016*; *Weiner et al., 2016*; *Wu et al., 2016*; *Keser et al., 2016*; *Kay and Yeatman, 2017*; *Lee Masson et al., 2017*; *Budisavljevic et al., 2018*; *Oishi et al., 2018*; *Panesar et al., 2019*; *Schurr et al., 2019*; *Broce et al., 2019*; *Jitsuishi et al., 2020*) as well as in non-human primate brains (macaque, *Takemura et al., 2017*; *Mars et al., 2018*; *Sani et al., 2019*; *Warrington et al., 2020*; vervet, *Sarubbo et al., 2019*; marmoset, *Kaneko et al., 2020*). Still, a precise definition of the VOF and neighboring pathways is hotly contested (*Bartsch et al., 2013*; *Catani et al., 2017*; *Martino and García-Porrero, 2013*; *Weiner et al., 2017*; *Panesar et al., 2019*; *Schurr et al., 2019*). Therefore, it is essential to investigate VOF anatomy in detail resolving debates among investigators and clarifying functional organization of the primate visual system.

3D-PLI data support the existence of the VOF in both coronal and sagittal sections. The VOF is located immediately lateral to the SS and directly connects dorsal and ventral extrastriate cortex as a distinct fiber bundle (*Figures 3C*, *4A–B,D* and *Figures 7–8*; *Figure 7—figure supplement 1*). While we cannot not rule out the possibility that a small number of axons may leave the VOF and take a rostro-caudal direction, our observation in 3D-PLI data supports the view that VOF fibers primarily travel along the superior-inferior axis and connect dorsal and ventral extrastriate cortex. Therefore, the 3D-PLI data validate an evidence of the VOF reported in dMRI using an independent measurement method with higher spatial precision and supports the hypothesis that dorsal and ventral extrastriate areas exchange information throughout VOF, rather than hypothesis on a large separation of anatomical pathways between dorsal and ventral visual areas.

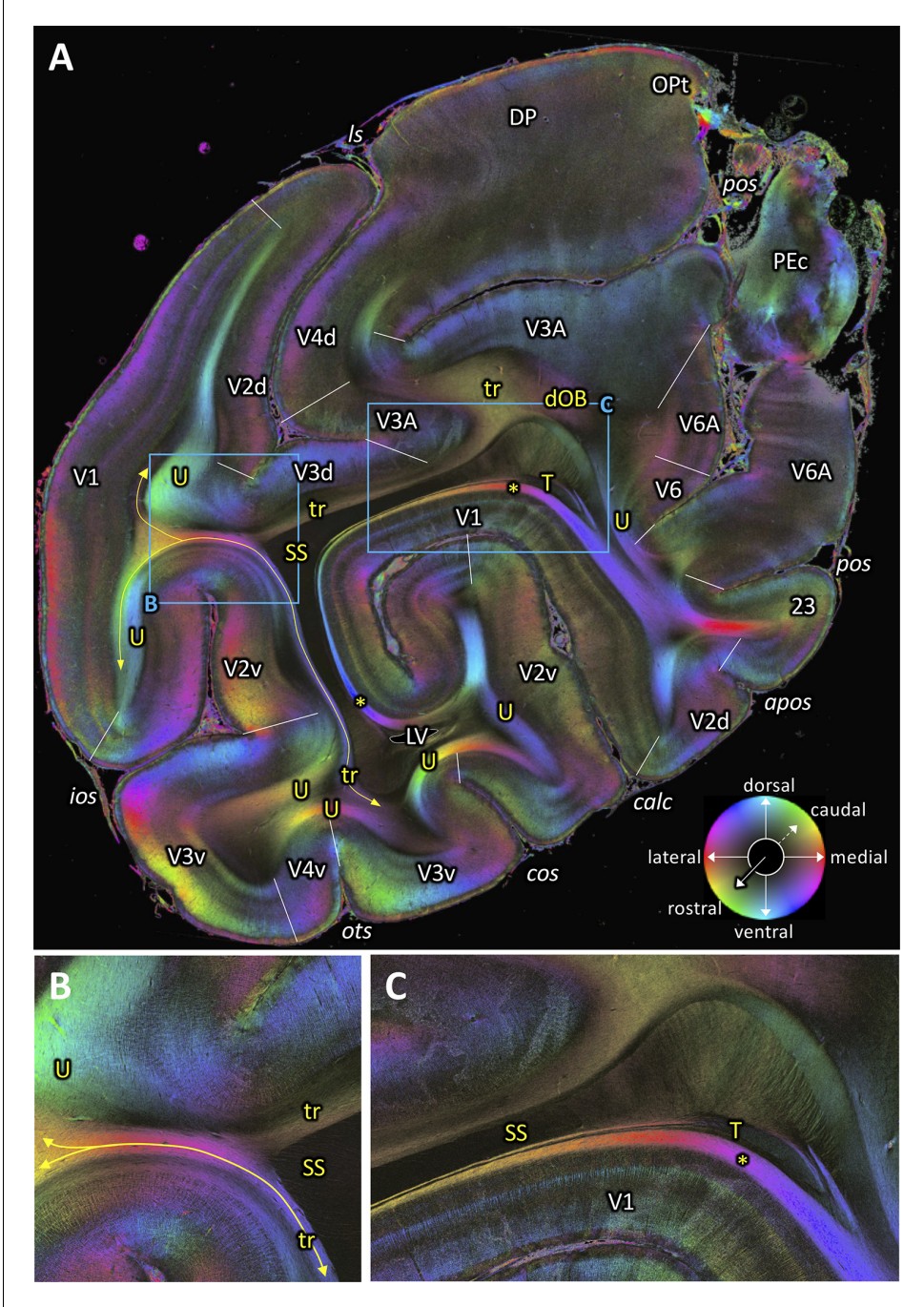

**Figure 9.** Fiber orientation map of a coronal section (brain ID1818; section #961, right hemisphere, caudal to that shown in *Figure 8*). (A) Overview with fiber tracts and cortical areas. (B-C) Magnifications of the respective rectangles in A. The sphere indicates the color coding of 3D fiber orientation in each pixel. 23: posterior cingulate area 23, apos: accessory parieto-occipital sulcus, calc: calcarine sulcus, cos: collateral sulcus, dOB: dorsal occipital bundle, DP: dorsal prelunate area, ios: inferior occipital sulcus, ls: lunate sulcus, LV: lateral ventricle, OPt: caudal inferior parietal lobule area, ots: occipito-temporal sulcus, PEc: superior parietal lobule caudal PE, pos: parieto-occipital sulcus, SS: stratum sagittale, T: Tapetum, tr: transverse fibers, U: U-fibers, V1: primary visual cortex, V2d: secondary visual cortex dorsal part, V2v: secondary visual cortex ventral part, V3A: visual area V3A, V3d: visual area 3 dorsal part, V3v: visual area 3 ventral part, V4d: visual area 4 dorsal part, V4v: visual area 4 ventral part, V6: visual area 6, V6A: visual area V6A. Asterisk indicates stratum calcarinum. Yellow lines with double arrow in (A and B) indicate a fiber tract between V1 and V3v.

*Figure 9 continued on next page*

*Figure 9 continued*

The online version of this article includes the following figure supplement(s) for figure 9:

**Figure supplement 1.** Part of the coronal section #961 (brain ID1818) from *Figure 9* showing the intricate spatial relations of the stratum calcarinum (*), tapetum (T), lateral ventricle (LV), ependyma (ep), and stratum sagittale (SS).

In macaques, VOF is adjacent to the ILF (*Schmahmann and Pandya, 2006*; *Takemura et al., 2017*). This makes it challenging to distinguish VOF from the ILF using dMRI-based tractography (*Takemura et al., 2017*). In FOM images of sagittal sections, we indeed found evidence that the VOF fibers merge into the inferior portion of the ILF (*Figure 4D*). However, in more lateral sagittal sections, the superior part of the ILF is completely separated from the VOF by a dark area between them (*Figure 4B*). It is also important to note that VOF fibers run following a superior-inferior direction immediately lateral to the SS and directly connect dorsal and ventral extrastriate cortex (*Figure 3C*). Therefore, fiber orientation of the VOF is fully distinct from the definition of the ILF, in which fibers run along the anterior-posterior axis and connect inferotemporal and occipital cortex. Thus, while VOF and ILF fibers merge and share some cortical terminations at the level of the ventral extrastriate cortex, we conclude that the VOF and the ILF are distinct bundles in the vervet monkey.

In coronal 3D-PLI data, we found that VOF can be divided into two distinct branches (medial and lateral branch), separated by the superior temporal sulcus (*Figure 7*). Since the medial branch connects parietal cortex and ventral occipitotemporal cortex, this branch is consistent with findings in previous papers investigating parieto-temporal connections using tracer-based autoradiography (*Seltzer and Pandya, 1984*; *Cavada and Goldman-Rakic, 1989*; *Schmahmann and Pandya, 2006*) or dMRI-based tractography in macaque monkeys (*Sani et al., 2019*). *Schmahmann and Pandya, 2006* interpreted this fiber bundle as a vertical limb of the ILF. However, since FOM data suggest that this medial branch is distinct from the neighboring pathways (*Figure 7D*) and merges into the VOF, it is questionable whether this fiber pathway should be seen as a part of the ILF system. Moreover, in the human dissection and dMRI literature, the parieto-temporal connection has been often distinguished from VOF and ILF, and identified as being the posterior arcuate fascicle (*Catani et al., 2005*; *Weiner et al., 2017*; *Panesar et al., 2019*; *Bullock et al., 2019*; *Schurr et al., 2019*). Similarities and differences of parieto-temporal connections between human and non-human primates should be a topic to be studied in future 3D-PLI works.

3D-PLI data also provide more precise evidence on the cortical termination of VOF by directly visualizing fibers running into the cortical gray matter (*Figure 7B*). While this type of analysis has been often used in dMRI studies (*Takemura et al., 2016*; *Takemura et al., 2017*), it is difficult to precisely measure fiber orientation at the border between gray and white matter at the resolution provided by dMRI and, therefore, it is not possible to make definitive statements concerning cortical fiber termination (*Reveley et al., 2015*). 3D-PLI data could directly visualize fibers terminating at dorsal extrastriate areas, such as V3d, V3A, V4d, V4t and DP (*Figures 4A–B*, *7A–B* and *8A*). While this is largely consistent with estimations from dMRI (*Takemura et al., 2017*), the present results provide concrete evidence regarding VOF cortical termination and thus validate findings from previous dMRI works. However, we note that even at PLI resolution, it is still difficult to precisely track the ventral termination of VOF because of the presence of U-fibers near OTS (*Figure 7C*). The challenge of disentangling a termination of a major bundle and a superficial U-fiber has been discussed in a previous dMRI work (*Reveley et al., 2015*). Resolving this challenge requires future extension of current 3D-PLI analysis framework in order to reconstruct three-dimensional fiber orientation distributions with high resolution and thus enable to track fiber orientation crosses with superficial U-fibers (*Axer et al., 2016*; *Schmitz et al., 2018*).

Despite existing limitations, fiber organization revealed by 3D-PLI data support the theory that VOF is important to transmit visual field information (*Takemura et al., 2016*; *Takemura et al., 2017*). Posterior coronal slices do not support the existence of a direct connection between dorsal and ventral extrastriate cortex, but rather support the existence of transverse fascicles connecting lateral V1 to dorsal or ventral extrastriate cortex (*Figure 9A–B*). At this stage, dorsal (V2d/V3d) and ventral extrastriate areas (V2v/V3v) have distinct visual field representations (dorsal: lower visual field, ventral: upper visual field; *Kolster et al., 2014*). The existence of a direct connection, VOF, becomes evident in more anterior slices, with extrastriate areas anterior to V3 containing complete

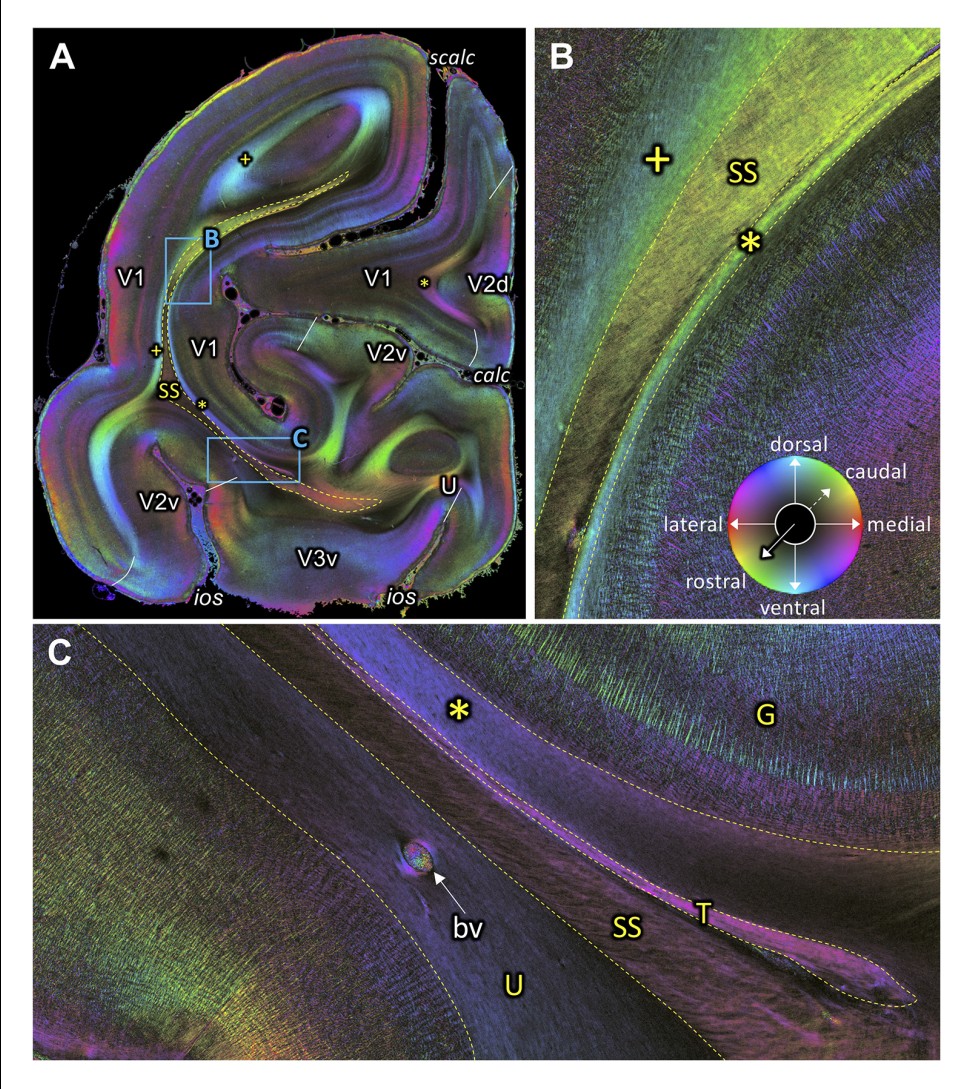

**Figure 10.** Fiber orientation map of a coronal section (brain ID1818; section #1061, right hemisphere, caudal to that shown in *Figure 9*). (A) Overview with fiber tracts and cortical areas. (B-C) Magnifications of the respective rectangles in A. The sphere indicates the color coding of 3D fiber orientation in each pixel. bv: blood vessel, calc: calcarine sulcus, G: Gennari stripe, ios: inferior occipital sulcus, scalc: superior calcarine sulcus, SS: stratum sagittale, T: tapetum, U: U-fibers, V1: primary visual cortex, V2d: secondary visual cortex dorsal part, V2v: secondary visual cortex ventral part, V3v: visual area three ventral part, +: fibers underlying lateral V1 (stratum extra-calcarinum). Asterisk indicates stratum calcarinum.

hemifield representations (*Figures 7–8*; *Kolster et al., 2014*; *Arcaro and Livingstone, 2017*; *Zhu and Vanduffel, 2019*). These results, together with more precise information on VOF cortical termination with 3D-PLI data, further suggest that VOF is involved in the integration of upper and lower visual field information in extrastriate areas with complete hemifield representations (*Takemura et al., 2016*; *Rokem et al., 2017*).

## Tapetum

While a number of neuroanatomists in the 19th century reported the existence of the tapetum, there has been substantial confusion regarding whether this fiber bundle should be considered as an association fiber or a callosal fiber (see *Schmahmann and Pandya, 2006*; *Schmahmann and Pandya, 2007*; *Forkel et al., 2015* for historical debates on the tapetum). Among classical neuroanatomists, *Burdach, 1822* reported that the tapetum fibers are an extension of the splenium corpus callosi.

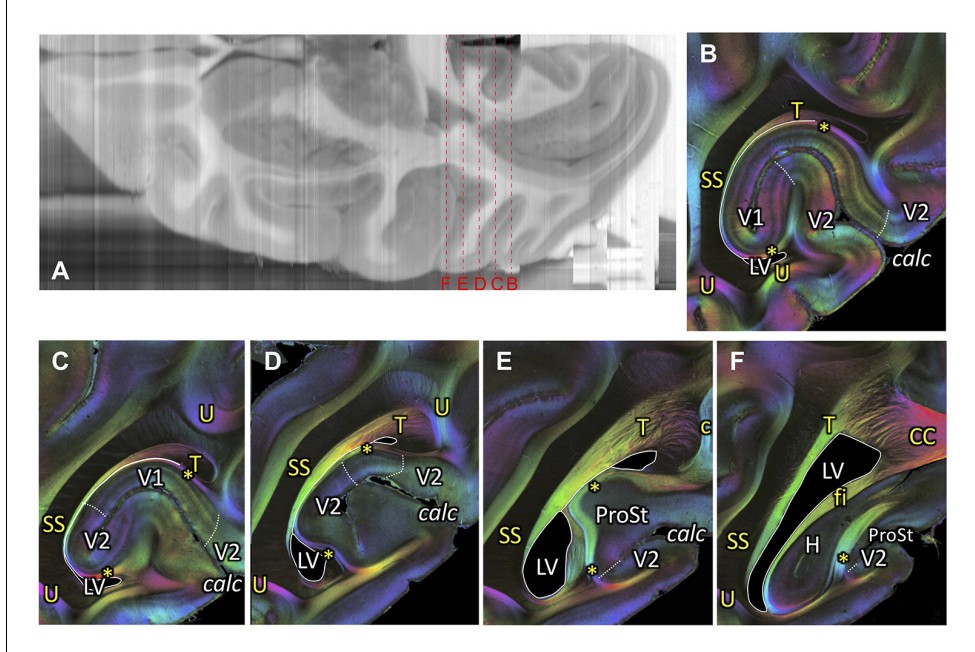

**Figure 11.** Course of the tapetum. (**A**) A reconstructed axial section of the right hemisphere from the coronal blockface images obtained during sectioning of vervet monkey brain ID1818. Red dashed lines and letters show the positions of the coronal sections in panels (**B**) (section #880), (**C**) (section #850), (**D**) (section #820), (**E**) (section #790), and (**F**) (section #759). (**B-F**) A series of FOM images of coronal sections from posterior (**B**) to anterior (**F**) indicate the tapetum (**B-F**), stratum calcarinum (**B-F**), SS (**B-F**), cingulum (**E**), and splenium fibers of the corpus callosum (**F**). In posterior slices, tapetum fibers are found between the SS and the primary visual cortex. In more anterior slices, the tapetum fibers merge with the splenium corporis callosi (**F**), demonstrating that the tapetum fibers are a continuation of corpus callosum fibers. c: cingulum, calc: calcarine sulcus, CC: corpus callosum, fi: fimbria hippocampi, H: hippocampal formation, LV: lateral ventricle, ProSt: area prostriata, SS: stratum sagittale, T: tapetum, U: U-fibers, V1: primary visual cortex, V2: secondary visual cortex. Asterisks indicate stratum calcarinum.

Since this observation was later supported by a number of investigators using various methods, callosal origin of tapetum fibers may no longer be debated (*Mettler, 1935*; *Clarke and Miklossy, 1990*; see *Schmahmann and Pandya, 2006* for a review). 3D-PLI data indeed directly visualized that tapetum fibers continue into splenium corpus callosI (*Figure 11*), confirming these previous works. 3D-PLI data further demonstrated the detailed course of tapetum fibers, namely its relative course with respect to neighboring fasciculi (SS and stratum calcarinum, as discussed below), lateral ventricle and ependyma (*Figures 9C*, *10C* and *11* and *Figure 9—figure supplement 1*). This detailed information will provide essential insights for guiding dMRI-based tractography studies on splenium fibers, which have been considered to be relevant for important cortical functions such as reading (*Binder and Mohr, 1992*; *Dougherty et al., 2007*).

## Short-range fiber systems in 3D-PLI data and classical dissection studies

Classical dissection studies by Sachs and Déjerine reported a number of intrinsic, short-distance fiber systems in the human occipital lobe (*Sachs, 1892*; *Déjerine, 1895*; *Vergani et al., 2014*; *Forkel et al., 2015*). We found a number of short-distance, U-fiber systems, which correspond to the descriptions in these classical works.

*Sachs, 1892* described that the 'stratum calcarinum' consists of fibers that circumvent the calcarine sulcus, the longest of which connects the cuneus to the lingual gyrus. *Déjerine, 1895* described that the stratum calcarinum is a U-fiber layer of the calcarine sulcus and connects the superior lip of the calcarine sulcus to its inferior lip. Consistent with these observations, we found a thick U-fiber layer surrounding the calcarine sulcus in the coronal series of slices (*Figures 7–9*). Importantly, the stratum calcarinum is distinct from the tapetum such that stratum calcarinum is an inner layer of U-fiber of the calcarine sulcus, whereas the tapetum is an outer layer (*Figure 9C*).

Sachs and Déjerine also described the occipital transverse fascicle of the cuneus (the transverse stratum of the cuneus of Sachs). According to *Déjerine, 1895*, this fascicle 'connects the cuneus to the convexity of the occipital lobe and to its inferior lateral aspect'. *Schmahmann and Pandya, 2006* interpreted that the dOB in macaque is homologous to this pathway, since it connects medial and lateral aspects of the dorsal occipital lobe. Consistent with observation by *Schmahmann and Pandya, 2006*, we also found that a transverse fascicle, dOB, connects lateral and medial portions of the dorsal occipital cortex (*Figure 8A*).

*Sachs, 1892* also reported a number of U-fibers along sulci in the human occipital lobe. Since the vervet monkey and human occipital lobe differ in their sulcal pattern, it is not possible to directly compare all U-fibers in Sachs atlas and present 3D-PLI data. However, one of the prominent U-fibers reported by *Sachs, 1892* is the stratum proprium sulci collateralis, which is a U-fiber of the collateral sulcus and connects areas of the lingual and fusiform gyri. In coronal FOM images, we consistently found U-fibers along the collateral sulcus (*Figures 7–9*), consistent with the description by *Sachs, 1892*. This fiber may carry information among ventral V2, V3 and V4.

While a number of bundles reported in the classical atlases by Sachs and Déjerine can be identified in 3D-PLI images, we could not find a fiber bundle similar to the 'occipital transverse fascicle of the lingual lobule of Vialet'. This is a fiber bundle identified by *Vialet, 1893* using myelin-stained material (*Vialet, 1893*). According to *Déjerine, 1895*, this fiber bundle connects the inferior lip of the calcarine sulcus to the convexity of the hemisphere. However, we do not find clear evidence for an uninterrupted fascicle directly connecting inferior lip of the calcarine sulcus to the lateral side of inferior occipital lobe. Rather, we only found U-fibers along sulci in the ventral occipital lobe, such as the stratum proprium sulci, or a branch of the VOF. This discrepancy may be explained by the fact that classical work mislabeled U-fibers as transverse fascicles, or that the larger human brain has a more distinct transverse fascicle in the ventral occipital cortex, which is less evident in vervet monkey. To test this hypothesis, future studies on human 3D-PLI data are required.

While there are a number of consistencies between classical dissection works in humans and 3D-PLI data in vervet monkeys, this study provides evidence of several pathways which have not been well described in previous works. We found that there are two transverse fascicles adjacent to the SS, in posterior sections where VOF is no longer visible (*Figure 9A–B*). These fascicles may have a role in carrying information between the lateral part of V1 and dorsal or ventral extrastriate areas. In further posterior sections, we found another fiber bundle, the stratum extra-calcarinum, directly underlying the lateral part of V1 (*Figures 4C,G* and *10A–B*). Since the lateral portion of V1 is involved in foveal information processing, characterization of these fibers is essential to understand mid-level cortical processing of foveal visual information, which is essential for object and face processing in the infero-temporal cortex.

Along the fundus of the superior temporal sulcus we found fibers running between the VOF and the cortical ribbon (hashtag symbol in *Figure 7* and *Figure 7—figure supplement 1*). While this fiber bundle has a similar fiber orientation to that of the VOF, it is clearly distinguishable from the VOF in FOM images (*Figure 7D*). This fiber bundle is consistent in location with a fiber bundle visible in tracer data of *Schmahmann and Pandya, 2006*'s work (*Figure 7—figure supplement 2*), but it has not yet been well characterized as a distinct bundle. It is likely that this fiber bundle plays an essential role in cortical inputs to or outputs from visual motion-selective areas MT or MST, which are located in the fundus of the superior temporal sulcus and may receive projections via this bundle.

## Vervet monkey as a non-human primate model for neuroscience studies

In this study, we investigated the organization of fiber tracts in the visual system of vervet monkeys (*Chlorocebus aethiops sabaeus*). While historically the macaque monkeys (*Macaca mulatta*) have been widely tested in visual neuroscience studies, vervet monkeys became an increasingly important model for neuroscience studies because of its biosafety (*Baulu et al., 2002*), lower cost (*Freimer et al., 2008*) and similarity of age-related diseases with those of humans (*Cramer et al., 2018*; *Latimer et al., 2019*). In fact, there is an increasing number of neuroscience studies investigating vervet monkeys as a non-human primate model, including studies investigating fiber tracts (*Fears et al., 2009*; *Fears et al., 2011*; *Woods et al., 2011*; *Fedorov et al., 2011*; *Lundell et al., 2011*; *Dyrby et al., 2013*; *Dyrby et al., 2014*; *Maldjian et al., 2014*; *Donahue et al., 2016*; *Menzel et al., 2019*; *Sarubbo et al., 2019*; *Barrett et al., 2020*). We also note that while vervet

monkeys are no more closely related to humans than are macaque monkeys, they are no more distantly related to humans than macaques. We think that the diversity of non-human primate model species will help to assure that the elucidated fiber tracts are representative more broadly of old world monkeys, not just a unique feature of a single species. Furthermore, our receptor autoradiography data also suggest cortical area of the visual system is broadly similar between vervet and macaque monkeys (*Figure 12*; *Figure 12—figure supplement 1*). Therefore, present investigation of vervet monkey visual system using 3D-PLI is an essential step toward the understanding of the organization of the primate visual system.

## Role of 3D-PLI for studying structural connectivity of the cerebral cortex

Over the last century, a number of invasive methods has been developed for analyzing cortico-cortical connections through white matter, from Klingler's dissection, strychnine neuronography, Nauta method, and modern tract tracing methods (*Lanciego and Wouterlood, 2011*; *Lanciego and Wouterlood, 2020*; *Takemura et al., 2019b*). Among these methods, axonal tract tracing has been the most widely used to study the visual system of non-human primates because of the high specificity and greater confidence in identifying cell bodies of neurons which are origins or terminations of axonal connections (*Kennedy et al., 2013*; *Rockland, 2020*), and the course of fiber tracts through the white matter was analyzed by means of radiolabeled isotopes (*Schmahmann and Pandya, 2006*) or MR-visible tracers (*Saleem et al., 2002*). Still, the localization of all visual fiber tracts of the white matter cannot be completely revealed by this technique, because the degree of complete visualization of the fiber tracts depends on the selection of injection sites and the amount of injected tracers. A fiber tract can contain axonal projections from different sites of origin and termination. Restricted injections of tracers may, therefore, label only discrete portions of a fiber tract. Moreover, axonal tracing does not provide direct visualization of fiber orientation in the white matter, and thus it is difficult to understand spatial organization of fiber bundles solely from tracer data. Thus, axonal tracing is an excellent approach to identify the projections from and to a well-defined cortical region, but cannot visualize all visual fiber tracts in the white matter.

More recently, there are resurgent interests in studying properties of white matter tracts in the human visual system using dMRI and tractography, which can demonstrate the position and trajectories of large and expected connections in living brains (*Mori et al., 1999*; *Conturo et al., 1999*; *Catani et al., 2002*; *Behrens et al., 2003*; *Wakana et al., 2004*; *Sherbondy et al., 2008*; *Rokem et al., 2017*). A major advantage of dMRI is its applicability in living brains, and it has provided valuable insights into the structural connections of the brain (*Catani and Thiebaut de Schotten, 2012*; *Li et al., 2013*; *Wandell, 2016*). dMRI studies have raised many important questions for visual neuroscience, such as the role of the historically neglected white matter tract connecting the dorsal and ventral visual streams (*Yeatman et al., 2014*; *Takemura et al., 2016*), white matter impairments in clinical disorders (*Ogawa et al., 2014*; *Takemura et al., 2019a*), and relationship between white matter properties and perceptual performance (*Thiebaut de Schotten et al., 2011*; *Rokem et al., 2017*). However, since this method has some limitations in disentangling of fiber pathways crossing within a voxel (*Jbabdi and Johansen-Berg, 2011*; *Assaf et al., 2019*; *Maier-Hein et al., 2017*), dMRI-based tractography requires accurate anatomical prior information to identify white matter tracts without producing false positives (*Catani et al., 2002*; *Wakana et al., 2004*; *Takemura et al., 2019b*). Currently, dMRI-based tractography relies on anatomical knowledge of Klingler's dissection studies (*Catani et al., 2002*; *Catani and Thiebaut de Schotten, 2012*) and axonal tract tracing (*Schmahmann and Pandya, 2006*), which are suited for describing the approximate position and trajectory or the origin and termination of axons belonging to white matter tracts, respectively. Thus, in order to improve our current ability to unambiguously identify fiber tracts from dMRI data, improved prior anatomical knowledge is required.

There are large gaps between tracer and dMRI regarding advantages and limitations. First, while tracers make is possible to label the origin or termination of fibers at a level of single pyramidal neurons (micrometer scale), dMRI has a much coarser resolution (millimeter scale). Second, dMRI is applicable for studying humans, but tracers are only applicable to non-human primates. Third, as discussed above, tracers have a dependency on the selection of injection sites, whereas dMRI provides three-dimensional data covering the whole brain. Although a number of studies attempted to directly compare results between tracer and dMRI analyses (*Thomas et al., 2014*; *Azadbakht et al.,*

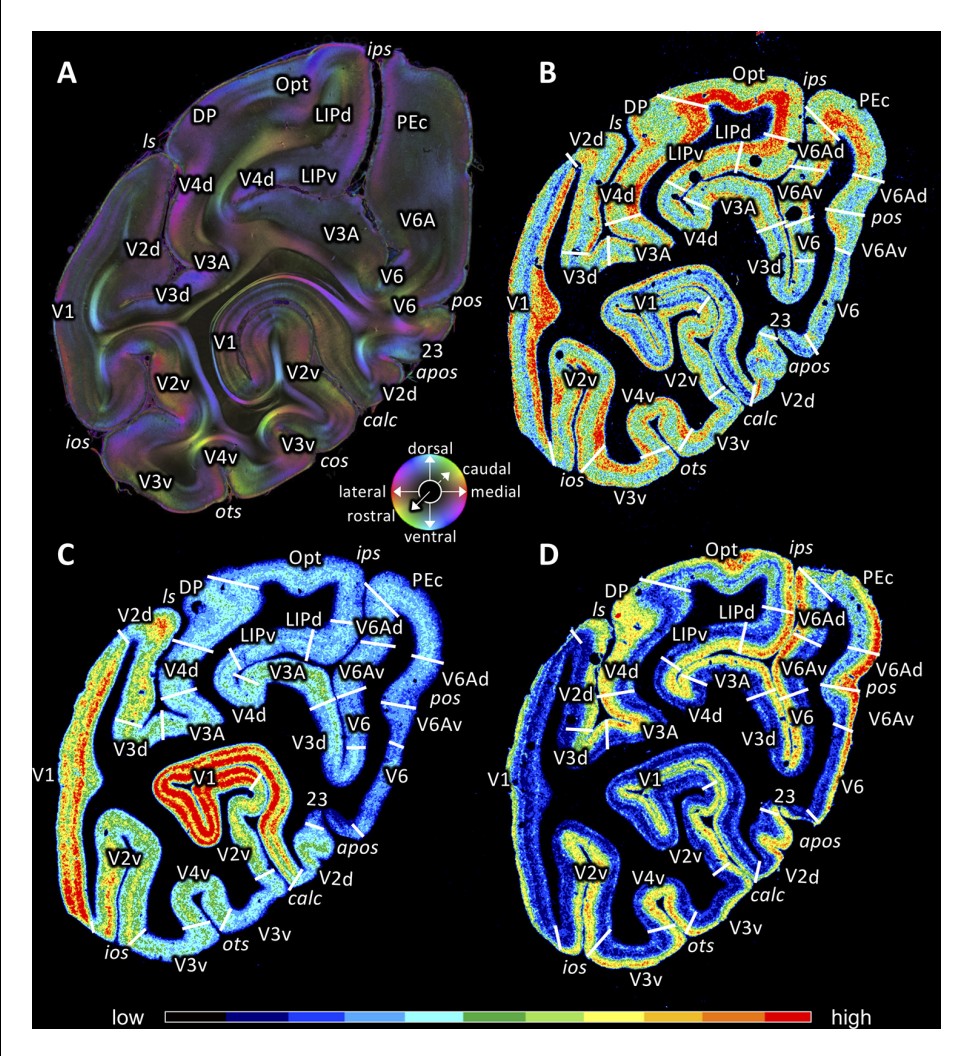

**Figure 12.** Coronal sections through the occipital lobe of vervet monkey brain ID1818 (**A** section #940) and vervet monkey brain ID1695 (**B-D**) depicting fiber orientation map (FOM; **A**) and the distributions of the glutamate kainate receptor (**B** section #761), the cholinergic muscarinic M$_2$ receptor (**C** section #768) and the noradrenergic α$_1$ receptor (**D** section #772). Cortical areas in (**A**) were identified by comparison with (**B-D**). The sphere indicates the color coding of 3D fiber orientation in each pixel in panel **A**. Receptor densities in **B-D** are color coded according to the scale at the bottom. 23: posterior cingulate area 23, apos: accessory parieto-occipital sulcus, calc: calcarine sulcus, cos: collateral sulcus, DP: dorsal prelunate area, ios: inferior occipital sulcus, ips: intraparietal sulcus, LIPd: dorsal part of the lateral intraparietal area LIP, LIPv: ventral part of the lateral intraparietal area LIP, ls: lunate sulcus, OPt: caudal inferior parietal lobule area, ots: occipito-temporal sulcus, PEc: superior parietal lobule caudal PE, pos: parieto-occipital sulcus, V1: primary visual cortex, V2d: secondary visual cortex dorsal part, V2v: secondary visual cortex ventral part, V3A: visual area V3A, V3d: visual area three dorsal part, V3v: visual area three ventral part, V4d: visual area four dorsal part, V4v: visual area four ventral part, V6: visual area 6, V6A: visual area V6A, V6Ad: visual area V6A dorsal part, V6Av: visual area V6A ventral part.

The online version of this article includes the following figure supplement(s) for figure 12:

**Figure supplement 1.** Coronal sections through the occipital lobe of macaque monkey ID11539 depicting a myelin stain (**A**) and the distributions of the glutamate kainate receptor (**B**), the cholinergic muscarinic M$_2$ receptor (**C**) and the noradrenergic α$_1$ receptor (**D**).

*2015*; *van den Heuvel et al., 2015*; *Donahue et al., 2016*; *Aydogan et al., 2018*; *Schilling et al., 2019*), there is a great challenge to directly compare wiring diagrams proposed by tracer studies and visual white matter tracts demonstrated by dMRI.

3D-PLI has an essential role to fill an inherent gap between tracer and dMRI studies, because 3D-PLI provides a similar data format to that of dMRI (FOM maps), though with a much higher spatial precision. In this study, we could demonstrate that 3D-PLI data reveal the cortical areas into which VOF fibers project (*Figures 4A–B*, *7A–B* and *8A*). Although it is possible to estimate putative cortical endpoint of the VOF from dMRI (*Takemura et al., 2016*; *Takemura et al., 2017*), such an estimation is known to be prone to biases due to a limitation of the measurements (*Reveley et al., 2015*). Therefore, 3D-PLI has a strong advantage to fill a gap between the knowledge of cortico-cortical connections from tracer studies and knowledge of fiber tract courses from dMRI or Klingler's dissection studies.

We also note that 3D-PLI approach has a strong advantage in its wide field of view, which enables the visualization of fiber orientation in the whole section of human or non-human primate brains, while other approaches, such as polarization sensitive optical coherence tomography (*Wang et al., 2014*; *Wang et al., 2018*) has a limited field of view. Therefore, to address a question at the level of whole single fiber tract, 3D-PLI is a well suitable approach because the dataset provides the visualization of whole single fiber tract, such as VOF or ILF.

While 3D-PLI is one of the most powerful histological methods for mapping nerve fiber bundles, polarimetric measurements (i.e., using one defined direction of light passage) as performed for this study and the corresponding analysis based on the most basic effective physical model, inherently pose some limitations, which need consideration here. 3D-PLI yields only a single fiber orientation for one measured tissue voxel, even if it is composed of crossing fibers with different fiber orientations. For a voxel size in this study (1.3 µm x 1.3 µm x 60 µm), a few tens of fibers might occupy a voxel, assuming fiber diameters between 0.4 and 15 µm according to *Aboitiz et al., 1992*. This number of fibers per voxel holds true for (dense) white matter fiber bundles, but is much smaller in gray matter tissue. As demonstrated by *Dohmen et al., 2015* and *Menzel et al., 2015*, the derived fiber orientation vector (and its level of confidence) from 3D-PLI measurements significantly depends on the complexity (e.g., dispersion) of the underlying fiber population within a voxel. Brain regions with in-plane crossing fibers, for example, are characterized by small measured amplitudes/signals due to destructive superposition of birefringence effects. This situation cannot be differentiated from small amplitudes caused by low myelin density or by fibers pointing out of the sectioning-plane without additional efforts, such as (i) introducing advanced simulation approaches to support signal interpretation as proposed recently by *Menzel et al., 2020*, (ii) extending the polarizing microscope with oblique measurement feature (*Schmitz et al., 2018*), (iii) investigating different brains cut along different planes (as done in the present study), or (iv) reconstructing the original shape and anatomical structures by means of non-linearly aligning serial sections (*Ali et al., 2018*). The latter issue addresses one of the major challenges for modern neuroanatomy studies based on a large series of individual histological sections, and will continue to be an essential subject of research. For 3D-PLI, a precise volume reconstruction from serial sections is also crucial to enable fiber tractography across sections, determine microstructural characteristics of long- and short-range connections, and, ultimately become a reliable correlate to dMRI approaches. As a final note, the fact that crossings of fiber bundles at the meso-scale might lead to signal cancellation (and therefore to misinterpretation of orientation) can also be used in a beneficial way for neuroanatomical studies, including the delineation and manual tracing of pathways in individual and across sections. Such cancellations often appear as a few-pixel-wide dark band, clearly indicating the zone of (close to) 90° crossings with the same fraction of fibers projecting in each direction.

While we focus on the visual pathways in the vervet monkey brain in this work, the same approach will be generally applicable for other pathways or other systems. We believe that 3D-PLI will provide further venues for filling the gap between tracer and dMRI studies, and improving anatomical prior information for guiding dMRI-based tractography. Therefore, an extension of this work will continue to improve the accuracy and interpretation of non-invasive human neuroimaging work, which is applicable to a wide range of clinical or neuroscientific questions.

## Materials and methods

### Brain tissue

Vervet monkeys (*Chlorocebus aethiops sabaeus*) used in this study were part of the Vervet Research Colony and were housed at the Wake Forest School of Medicine. Macaque monkeys (*Macaca fascicularis*) were obtained from Covance (Münster, Germany). Animals were colony-born, of known age and were mother-reared in species-typical social groups. The present study did not include experimental procedures with live animals. Brains were obtained when animals were sacrificed to reduce the size of the colony, where they were maintained in accordance with the guidelines of the Directive 2010/63/eu of the European Parliament and of the Council on the protection of animals used for scientific purposes or the Wake Forest Institutional Animal Care and Use Committee IACUC #A11-219. Euthanasia procedures conformed to the AVMA Guidelines for the Euthanasia of Animals.

Two brains of vervet monkeys (monkey 1818, male, 2.4 years old, monkey 1947, 1 years old; both male) were removed from the skull after flush with phosphate buffered saline and perfusion fixation with 4% paraformaldehyde. The brains were immersed in 20% glycerin, deep frozen and stored at −70 C. The brain of one monkey (monkey 1818) was sectioned coronally. The left hemisphere of the other monkey (monkey 1947) was sectioned sagittally. Serial sectioning was performed using a large-scale cryostat microtome (Poly-cut CM 3500, Leica, Germany) at 60 μm thickness. During sectioning, each blockface of the frozen brain or hemisphere was digitized with a CCD camera to obtain aligned and undistorted reference images.

A third vervet monkey brain (ID 1695; 3.9 years, male) was processed for receptor autoradiography (*Palomero-Gallagher and Zilles, 2018*; *Zilles and Palomero-Gallagher, 2017*) in order to enable the multimodal identification and definition of the borders between cortical areas by comparison with 3D-PLI images at comparable levels. Additionally, three macaque monkey brains (IDs 11530, 11539, 11543; 6 ± 1 years, male) were processed for receptor autoradiography, thus enabling demonstration of the comparability between cortical segregation patterns in the vervet brain and that of a widely used non-human primate model. The brains from these animals were not flushed or perfused. Rather, each hemisphere was separated into an anterior and a posterior block at the height of the most caudal portion of the central sulcus and shock frozen in isopentane at −40°C to −50°C. A part of receptor autoradiography data from these brain has been already analyzed in a previous work (*Niu et al., 2020*).

### Polarized light imaging: Image acquisition and processing

Microscopic imaging referred to as 3D-PLI (*Axer et al., 2011a*; *Axer et al., 2011b*) was performed with a polarimetric setup based on a Köhler illuminated (wavelength spectrum: 550 ± 5 nm) bright field microscope equipped with two polarizing filters and a movable specimen stage (LMP-1, Taorad GmbH) (*Reckfort et al., 2015*). The field of view of the monochrome CCD camera (QImaging Retiga 4000R) was 2.7 × 2.7 mm$^2$, providing an in-plane pixel resolution of 1.3 μm. Consequently, imaging of large-area whole brain sections required a tile-wise scanning with 0.75 mm (sagittal series) or 1 mm (coronal series) overlaps. During the measurement, linearly polarized light was applied to the unstained sections and the transmitted light intensity was sampled by means of a circular analyzer unit at nine (sagittal series) or eighteen (coronal series) vertical polarization planes covering 180° of rotation.

Subsequent image analysis was performed utilizing high-performance computing (HPC) algorithms efficiently running on the supercomputing facility JURECA at the Jülich Supercomputing Center, JSC, Forschungszentrum Jülich, Germany. Image analysis included (1) image calibration with flat image scans (i.e., images of the glass slides at a position without tissue), (2) tile stitching based on feature detection and matching, (3) tissue/background segmentation by means of seeded region growing, and (4) determination of multiple physical and anatomical parameters (e.g., birefringence strength, light scattering and extinction, fiber orientations) utilizing Jones calculus approach (*Jones, 1941*; *Axer et al., 2011a*; *Axer et al., 2011b*). These processing steps were implemented as an automated HPC workflow (*Amunts et al., 2014*).

Given the labor-extensive nature of 3D-PLI data acquisitions and computational demands on image analyses as described above, we performed data collection and analyses in following steps. Image acquisition was initially performed in every 20th section. Based on this data, we identified a

range of sections covering the tracts of interest (the SS, the ILF, and the VOF) in this study. We then obtained 3D-PLI data in between the sections measured in the first step. Therefore, the measurement did not cover the entire brain, but we analyzed a series of sections completely covering occipital white matter fiber tracts.

Two of the determined parameters are of particular interest for this study; transmittance and fiber orientation maps (FOM). The transmittance map represents the (pixel-wise) average light intensities obtained from the rotation measurements and mainly reflects the light extinction (i.e., absorption and scattering) of brain tissue. Strong sources of scattering and absorption, such as myelinated fibers, appear dark. The FOM is a representation of the regional 3D fiber orientations (described as unit vectors and composed of two angles referred to as direction and inclination angles; cf. *Herold et al., 2019*, for detailed description of FOM generation). It provides unprecedented fiber contrast in both cortical and white matter regions, together with region-specific fiber orientation information. All FOMs were specifically color-coded in a modified version of the H(ue)S(aturation)V (alue) color space, i.e. the HSV-black version, where brightness decreases with increasing inclination, staining the poles black at 90°. Thus, fiber orientations were encoded by hue, saturation and brightness values. Comparison with other color encoding schemes often used in dMRI studies has been described in previous publications (*Axer et al., 2011b*; *Henssen et al., 2019a*; *Henssen et al., 2019b*).The reference for a single vector representation is often not a single fiber, but all birefringent tissue compartments inside a volume element (voxel) contribute to the measured signals.

## Quantitative in vitro receptor autoradiography: experimental procedures and image acquisition and processing

Each of the frozen blocks was serially sectioned in the coronal plane (20 μm thickness) at −20°C using a cryostat microtome (CM 3050, Leica, Germany). Sections were thaw-mounted onto glass slides, and freeze-dried overnight. Adjacent sections were processed for histological staining of cell bodies (*Merker, 1983*) or myelin sheaths (*Gallyas, 1979*), or for quantitative in vitro receptor autoradiography following standardized protocols (*Palomero-Gallagher and Zilles, 2018*; *Zilles and Palomero-Gallagher, 2017*). The glutamatergic kainate receptor, cholinergic muscarinic $M_2$ receptor and noradrenergic $\alpha_1$ receptor were visualized by incubating neighboring sections in solutions of respective tritiated receptor ligands. Non-specific binding was determined in parallel binding assays in which sections were incubated with the tritiated ligand and an unlabeled displacer. All binding assays were preceded by a 20–30 min preincubation in the respective buffer.

The kainate receptor was labelled with [$^3$H]kainate (8 nM) in a 50 mM Tris-citrate buffer (pH 7.1) containing 10 mM Ca-acetate at 4°C for 45 min. The unlabeled displacer was kainate (100 μM). This main incubation was followed by a washing step with buffer (3 × 4 s) and two dips into 100 ml acetone containing 2.5 ml glutaraldehyde.

The cholinergic muscarinic $M_2$ receptor was labelled with [$^3$H]oxotremorine-M (0.8 nM) in a 20 mM Hepes-Tris buffer (pH 7.5) containing 10 mM $MgCl_2$ at 22°C for 60 min. The unlabeled displacer was carbachol (1 μM). The main incubation was terminated by a washing step with buffer (2 × 2 min) followed by a dip in distilled water at 4°C.

The noradrenergic $\alpha_1$ receptor was labelled with [$^3$H]prazosin (0.2 nM) in a 50 mM Tris-HCl buffer (pH 7.4) at 30°C for 45 min. The unlabeled displacer was phentolamine (10 μM). The main incubation was terminated by a washing step with buffer (2 × 5 min) followed by a dip in distilled water at 4°C.

Following binding assays, sections were dried in a stream of warm air, then co-exposed together with standards of known concentrations of radioactivity to tritium-sensitive films (Hyperfilm, Amersham) for 4 (kainate receptor) or 6 ($M_2$ and $\alpha_1$ receptors) weeks. After developing the films, autoradiographs were digitized with a CCD-camera (Axiocam MRm, Zeiss, Germany) and the image processing software Axiovision (Zeiss, Germany). A transformation curve indicating the relationship between grey values in the autoradiograph and concentrations of radioactivity in the tissue was computed for each receptor type using in-house-developed Matlab (The MathWorks, Natick, MA) scripts and the standards with known radioactivity concentrations. Autoradiographs were then subjected to linear contrast enhancement, color coding and median filtering to provide a clear visualization of the regional and laminar receptor distribution patterns.

## Identification of cortical areas

The borders of visual areas V1, V2, V3V, V4V, V4T, V5/MT, and MST as well as of intraparietal areas LIPd, LIPv, VIP, and MIP can be identified in the vervet monkey based on differences in their myeloarchitecture as revealed in both transmittance images and fiber orientation maps (*Zilles et al., 2016*). These areas, as well as the areas defined in the parietal and temporal lobes are supported by comparisons between the brains of vervet monkey 1818, processed for PLI, and of vervet monkey 1695, processed for histological stainings and in vitro receptor autoradiography (*Figure 12*). Although fiber orientation maps and autoradiographs were obtained from different animals, the position of cortical areas identified by differences in myeloarchitecture as revealed by PLI (*Figure 12A*) is comparable to that of areas revealed by receptor distribution patterns (*Figure 12B–D*). Our parcellation of the vervet cortical ribbon is also supported by parallelisms between the regional and laminar distribution patterns of transmitter receptors in the vervet brain and those of the same receptors in sections obtained from comparable rostro-caudal levels in the macaque monkey brain (*Figure 12—figure supplement 1*). Furthermore, areas identified in the macaque brain have also been included in the atlases of *Saleem and Logothetis, 2012* and *Paxinos et al., 2009*.

Given the correspondence of borders identified by analysis of PLI and receptor autoradiographic datasets in the vervet brain as well as the topological and receptor architectonical comparability of areas identified in the coronally sectioned vervet and macaque brains, cortical areas in the sagittally sectioned vervet monkey (ID 1947) were identified based on regional and laminar differences in the distribution of transmitter receptors in sagittal sections through the macaque brain (*Zilles and Clarke, 1997*) as well as by comparison with the atlases of *Saleem and Logothetis, 2012* and *Paxinos et al., 2009*.

We have made 3D-PLI data used for generating figures publicly available via the EBRAINS platform of the Human Brain Project in order to provide data unbiased to our border definitions. The data (*Axer et al., 2020*) is accessible via the following DOI:10.25493/AFR3-KDK.

## Acknowledgements

We gratefully acknowledge the computing time granted through JARA-HPC on the supercomputer JURECA at Forschungszentrum Jülich (FZJ), Germany. We thank David A Leopold, Frank Q Ye and Tim. B Dyrby for providing dMRI dataset from macaque and vervet monkeys. We also thank Sabine Wittschonnke for the specimen with Klingler's dissection of the Stratum Sagittale depicted in *Figure 1A*.

## Additional information

### Funding

| Funder | Grant reference number | Author |
| --- | --- | --- |
| Japan Society for the Promotion of Science | JP17H04684 | Hiromasa Takemura |
| Japan Society for the Promotion of Science | JP15J00412 | Hiromasa Takemura |
| Horizon 2020 - Research and Innovation Framework Programme | 785907 (HBP SGA2) | Markus Axer<br>Karl Zilles |
| National Institutes of Health | R01 MH092311 | Roger Woods |
| NIH | P40 grant OD010965 | Matthew J Jorgensen |

The funders had no role in study design, data collection and interpretation, or the decision to submit the work for publication.

### Author contributions

Hiromasa Takemura, Conceptualization, Formal analysis, Funding acquisition, Investigation, Writing - original draft, Writing - review and editing; Nicola Palomero-Gallagher, Conceptualization, Data

curation, Formal analysis, Investigation, Visualization, Writing - original draft, Writing - review and editing; Markus Axer, Data curation, Funding acquisition, Investigation, Visualization, Methodology, Writing - review and editing; David Gräßel, Data curation, Investigation, Visualization, Writing - review and editing; Matthew J Jorgensen, Resources, Funding acquisition, Writing - review and editing; Roger Woods, Conceptualization, Resources, Funding acquisition, Writing - review and editing; Karl Zilles, Conceptualization, Formal analysis, Supervision, Funding acquisition, Investigation, Methodology, Writing - original draft, Project administration, Writing - review and editing

## Author ORCIDs

Hiromasa Takemura  https://orcid.org/0000-0002-2096-2384
Nicola Palomero-Gallagher  https://orcid.org/0000-0003-4463-8578
David Gräßel  http://orcid.org/0000-0003-3228-8048
Matthew J Jorgensen  https://orcid.org/0000-0002-0977-6425

## Ethics

Animal experimentation: Vervet monkeys (Chlorocebus aethiops sabaeus) used in this study were part of the Vervet Research Colony and were housed at the Wake Forest School of Medicine. Macaque monkeys (Macaca fascicularis) were obtained from Covance (Münster, Germany). Animals were colony-born, of known age and were mother-reared in species-typical social groups. The present study did not include experimental procedures with live animals. Brains were obtained when animals were sacrificed to reduce the size of the colony, where they were maintained in accordance with the guidelines of the Directive 2010/63/eu of the European Parliament and of the Council on the protection of animals used for scientific purposes or the Wake Forest Institutional Animal Care and Use Committee IACUC #A11-219. Euthanasia procedures conformed to the AVMA Guidelines for the Euthanasia of Animals.

## Decision letter and Author response

Decision letter https://doi.org/10.7554/eLife.55444.sa1
Author response https://doi.org/10.7554/eLife.55444.sa2

# Additional files

## Supplementary files

• Transparent reporting form

## Data availability

Original data is publicly available via the EBRAINS platform of the Human Brain Project (Axer et al., 2020; https://doi.org/10.25493/AFR3-KDK).

The following dataset was generated:

| Author(s) | Year | Dataset title | Dataset URL | Database and Identifier |
|---|---|---|---|---|
| Axer M, Gräßel D, Palomero-Gallagher N, Takemura H, Jorgensen MJ, Woods R, Amunts K | 2020 | Images of the nerve fiber architecture at micrometer-resolution in the vervet monkey visual system | https://kg.ebrains.eu/search/instances/Dataset/79db19fa-41bd-4292-9a33-e0e79dc9a9aa | EBRAINS, 10.25493/AFR3-KDK |

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
