## [Decision Letter]

**Acceptance summary:**

This fascinating and incredibly detailed work leverages cutting-edge tools for mapping the trajectories of long range connections in the brain to provide a critical roadmap for the visual system in non-human primates. In doing so, this project is able to clearly bridge multiple independent observations from previous studies into a unified summary of these critical white matter pathways. In doing so this work both answers many outstanding questions about the organization of the visual system in vervets (thus providing insights into primates in general) and provides a valuable resource for future investigations into the visual pathways.

**Decision letter after peer review:**

Thank you for submitting your article "Anatomy of nerve fiber bundles at micrometer-resolution in the vervet monkey visual system" for consideration by *eLife*. Your article has been reviewed by three peer reviewers, including Timothy Verstynen as the Reviewing Editor and Reviewer #1, and the evaluation has been overseen by Timothy Behrens as the Senior Editor.

The reviewers have discussed the reviews with one another and the Reviewing Editor has drafted this decision to help you prepare a revised submission.

Summary:

Takemura and colleagues describe a detailed analysis of the trajectories of major white matter pathways of the visual system in the vervet monkey using postmortem polarized light imaging (PLI). The authors clearly outline the trajectories of both major and minor pathways, addressing controversies in the literature, to a level of detail that has yet to be done given the limitations of previous methods (e.g., diffusion imaging and Klinger dissection). The study is methodologically exceptional and highly detailed. It is a phenomenal resource paper that will be of significant value to the neuroanatomy and visual sciences fields.

All three reviewers felt that this was a very powerful and important study and were supportive of seeing this get published. There, however, some major concerns that need addressing before it is ready for acceptance. I would like to point out, in particular, the concerns about data availability (concerns that are also shared by the Reviewing and Senior Editors) are particularly serious given *eLife*'s commitment to open science practices.

Essential revisions:

1) Framing.

Reviewer #1 felt that the authors' framing of the motivation of this study, as addressing major "controversies" about the existence and trajectory of white matter pathways in the visual stream, might inflate the scope of the actual work. In many cases, such controversies about these pathways are minor or long since discarded by the neuroanatomy field. For example, the existence of the ILF was doubted in a single study dating back to the 1980s. Most major neuroanatomy textbooks, and the field at large, accept its existence and agree on the principle trajectories of the ILF. This framing puts a false sense of breadth on the importance of the study. Realistically, this study is an analysis of the major pathways of the vervet visual system as they present in the same animals. This is a substantial enough contribution to the literature without the need to inflate the scope of the work as addressing critical controversies that may not be present.

2) Links to prior work.

All three reviewers had concerns about the study's link to prior work and existing knowledge.

Reviewer #1 points out that, over the last 10 years, there has been a significant expansion of diffusion imaging analysis of major white matter pathways that include many of the same pathways studied here (e.g., (Mori et al., 2008; Zhang et al., 2008; Catani and de Schotten, 2008; Yeh et al., 2018)), including focal analysis of the ventral visual pathways (e.g., (Pyles et al., 2013; Gschwind et al., 2012; Kamali et al., 2014; Toosy et al., 2004)). In order to put the value of the current work into perspective, the authors need to do a more comprehensive comparison of what is known in the literature and make distinctions clear with regards to species.

Along the same vein, reviewer #2 point out that the choice of vervet brain is somewhat puzzling. There is not nearly as much prior diffusion tractography or anatomical tract-tracing as on the rhesus macaque. And of course, it's imperfect compared to the human, and it is possible to do 3D-PLI on human brain (as some of the authors have, in fact, done). And vervets are not more closely related to humans than macaques. The reviewer's concern is that, rather than clarifying, this further muddies the waters (and limits impact), by adding a new species and a new method simultaneously. This should be directly addressed.

Reviewer #3 raises a similar point, highlighting that the areas of the visual cortex are identified with reference to atlases, but these are in the macaque, not the vervet monkey. This should be acknowledged. It would be nice to see a bit more discussion on the autoradiography and criteria for defining areas, as this is highly specialized knowledge that we have to take at face value. At least a reference to the various criteria for identifying areas would be helpful.

3) Summary of work.

Reviewers #1 and #2 both point out that the authors do a fantastic job of characterizing these pathways from multiple perspectives. Yet the proverbial forest gets lost for the trees (i.e., it is hard to keep track of the details across images and text descriptions). Both reviewers feel that a visual summary of the trajectory of all the major pathways identified here would go a long way to assisting readers in understanding the take home message of the paper. Reviewer #2 suggests something along the lines of the 3D representations from their 2016 paper, along with some black and white atlas-style coronal slices showing the boundaries of the different bundles. This reviewer pointed out that there are a number of potential readers who will not want to dive into the color schemes used in the current figures, but will want to know the placement of the bundles and general orientation of the fibers.

Reviewer #2 points out that an additional table summarizing entry/exit points for each of the various fiber bundles would be so useful. One of the advantages of this method over diffusion tractography is the improved ability to distinguish these at 1.3μm resolution, 3D-PLI approaches the width of a single axon. It is fine that some of these are unclear because of crossing U-fibers, that can be included in the table.

4) Clarity of findings.

Reviewer #2 asks how confident are the authors that the VOF is actually primarily bringing fibers from dorsal to ventral brain areas, and not bringing dorsal and ventral fibers to the middle to traverse rostral/caudally (say, in the ILF?).

Reviewer #3 points out that the authors might consider providing a bit more context in the Results section. Many of the tracts are discussed without any background and the logic of their decision to call a bundle by a certain name is left implicit. Surely, the authors have clear reasons to label their tracts, so it would be nice if some of this implicit-and no doubt enormous-reservoir of knowledge is shared. There is extensive review in the Discussion section, but the Results section could be made more readable. In general, more extensive reference to the literature would benefit the paper. For instance, there is repeated referenced to the 'controversy' around VOF, but this is supported by a citation to the paper that identified the tract-hardly a statement of controversy.

5) Data availability.

Reviewers #1 and 3, as well as the Senior Editor, feel that the data availability afforded by the authors does not meet the standards required by *eLife*. Reviewer #3 points out that the authors state that all data is presented in the figures, but of course that cannot even come close to demonstrating the richness of their data. In this era of open science we would very much hope they would create a resource for scientists based on their data; this would also be the only way that a reader can really evaluate their claims, as obviously the current manuscript by its nature is descriptive and reliant on the sections the authors choose to show. The reviewer appreciates that these data probably are so large as to make simple online sharing prohibitive, but the authors would be applauded for setting out a trajectory for sharing this type of data. At the very least they should rephrase their data availability statement.

---

## [Author Response]

Summary:Takemura and colleagues describe a detailed analysis of the trajectories of major white matter pathways of the visual system in the vervet monkey using postmortem polarized light imaging (PLI). The authors clearly outline the trajectories of both major and minor pathways, addressing controversies in the literature, to a level of detail that has yet to be done given the limitations of previous methods (e.g., diffusion imaging and Klinger dissection). The study is methodologically exceptional and highly detailed. It is a phenomenal resource paper that will be of significant value to the neuroanatomy and visual sciences fields.All three reviewers felt that this was a very powerful and important study and were supportive of seeing this get published. There, however, some major concerns that need addressing before it is ready for acceptance. I would like to point out, in particular, the concerns about data availability (concerns that are also shared by the Reviewing and Senior Editors) are particularly serious given eLife's commitment to open science practices.

We thank the editors and reviewers for the constructive comments and positive assessments of the scope of this paper. We address each point below.

Essential revisions:1) Framing.Reviewer #1 felt that the authors' framing of the motivation of this study, as addressing major "controversies" about the existence and trajectory of white matter pathways in the visual stream, might inflate the scope of the actual work. In many cases, such controversies about these pathways are minor or long since discarded by the neuroanatomy field. For example, the existence of the ILF was doubted in a single study dating back to the 1980s. Most major neuroanatomy textbooks, and the field at large, accept its existence and agree on the principle trajectories of the ILF. This framing puts a false sense of breadth on the importance of the study. Realistically, this study is an analysis of the major pathways of the vervet visual system as they present in the same animals. This is a substantial enough contribution to the literature without the need to inflate the scope of the work as addressing critical controversies that may not be present.

We thank the reviewer for this suggestion. We removed texts referring to “controversies” from the Abstract and Introduction in the revised manuscript, and clarified that a major goal of this study is to investigate the detailed spatial organization of fiber tracts in the vervet monkey visual system. We also agree with the reviewers that the majority of researchers no longer dispute the existence of the ILF. Therefore, we revised the text by stating that the existence of the ILF has been only doubted by a single study.

We also specified species when we describe the literature in the Introduction, following reviewer’s comment (see our response to questions on species below).

Specific changes: In the Abstract, we removed texts on controversies and reorganized texts as follows:

“Although the primate visual system has been extensively studied, detailed spatial organization of white matter fiber tracts carrying visual information between areas has not been fully established. This is mainly due to the large gap between tracer studies and diffusion MRI studies, which focus on specific axonal connections and macroscale organization of fiber tracts, respectively.”

In the Introduction, we also made an ample revision in a paragraph originally referring to controversies:

“Despite the collections of dissection, tracer and dMRI studies on the visual system, we do not fully understand detailed spatial organization of the white matter tracts in the visual system because there remains a large gap between studies performed by different methods (Takemura et al., 2019b; Rushmore et al., 2020). Specifically, while tracers are well suited to measure specific connections from or to injection sites, this method is not able to visualize the entire fiber tracts. On the other hand, while dMRI is well suited for measuring approximate position and trajectories of major fiber tracts, it does not have enough resolution to precisely measure termination of fiber tracts in cortical gray matter (Reveley et al., 2015). Therefore, there is a large gap between findings on cortico-cortical connectivity from tracer studies and findings on white matter tracts from dMRI studies. Moreover, there are many remaining questions regarding spatial organization of white matter tracts, since it is difficult to precisely measure such organization using any of the aforementioned methods. For example, it is not yet clear how much the vertical occipital fascicle (VOF; Yeatman et al., 2014) is an independent fascicle from the inferior longitudinal fascicle (ILF) in the macaque (Schmahmann and Pandya, 2006; Takemura et al., 2017). Moreover, the spatial organization of neighboring tracts, such as the stratum sagittale (SS) and the ILF, has been controversially discussed among investigators (Schmahmann and Pandya, 2006). We also note that not all studies reported the same fiber tracts or proposed identical definitions of fiber tracts (Schmahmann and Pandya, 2006; Yeatman et al., 2014). These ambiguities in the literature result partly from different methods used for each study (dissection, dMRI and tracer) because these methods have their own advantages and limitations. A study visualizing whole fiber bundles with higher spatial resolution seems necessary to fill a gap between different methods and to establish our understanding on the detailed spatial organization of visual white matter tracts.”

In the Discussion section, we wrote:

“Although in the past the existence of the ILF was questioned (Tusa and Ungerleider, 1985), later human dMRI-based tractography (Catani et al., 2003) and macaque tracer studies (Schmahmann and Pandya, 2006) have demonstrated the existence of the ILF as a longitudinal association fiber.”

2) Links to prior work.All three reviewers had concerns about the study's link to prior work and existing knowledge.Reviewer #1 points out that, over the last 10 years, there has been a significant expansion of diffusion imaging analysis of major white matter pathways that include many of the same pathways studied here (e.g., (Mori et al., 2008; Zhang et al., 2008; Catani and de Schotten, 2008; Yeh et al., 2018)), including focal analysis of the ventral visual pathways (e.g., (Pyles et al., 2013; Gschwind et al., 2012; Kamali et al., 2014; Toosy et al., 2004)). In order to put the value of the current work into perspective, the authors need to do a more comprehensive comparison of what is known in the literature and make distinctions clear with regards to species.

We thank the reviewer for pointing out these previous works. In the revised manuscript, we cited all of these previous publications and incorporated descriptions on these dMRI studies into the Introduction and Discussion section. We included a discussion on the ILF literature in the context of categorical information processing, by citing works proposed by reviewers and also from other groups.

We also revised the text of the Introduction, by removing our description on controversies and incorporating extensive discussion on what is known and what is unknown in the tracer and diffusion MRI literature.

Specific changes: In the Introduction, we included following texts:

“The advancement of dMRI acquisition and analysis methods led developments on the atlases of human major fiber tracts (Mori et al., 2008; Catani and Thiebaut de Schotten, 2008; 2012; Yeh et al., 2018) and automated procedures to analyze those tracts based on dMRI data (Zhang et al., 2008; Yendiki et al., 2011; Yeatman et al., 2012; 2018; Wassermann et al., 2016; Wasserthal et al., 2018; Warrington et al., 2020).”

“Although these tracts have already been identified (macaque studies, Schmahmann and Pandya, 2006; Takemura et al., 2017; human studies, Catani et al., 2003; Toosy et al., 2004; Catani and Thiebaut de Schotten, 2008; 2012; Kamali et al., 2014; Yeatman et al., 2014), these studies either did not provide direct evidence of the underlying anatomical structure, or are prone to methodical limitations (see Discussion).”

In the Discussion section, we wrote:

“Further analysis of human dMRI together with fMRI or behavioral data suggests a relevance of human ILF with categorical information processing in ventral visual stream (Gschwind et al., 2012; Pyles et al., 2013; Scherf et al., 2014; Tavor et al., 2014).”

Along the same vein, reviewer #2 point out that the choice of vervet brain is somewhat puzzling. There is not nearly as much prior diffusion tractography or anatomical tract-tracing as on the rhesus macaque. And of course, it's imperfect compared to the human, and it is possible to do 3D-PLI on human brain (as some of the authors have, in fact, done). And vervets are not more closely related to humans than macaques. The reviewer's concern is that, rather than clarifying, this further muddies the waters (and limits impact), by adding a new species and a new method simultaneously. This should be directly addressed.

As the reviewer pointed out, in principle, 3D-PLI can be applicable to the human brain (e.g. Zeineh et al., 2017). In practice, 3D-PLI measurements on human brains require significantly longer time than 3D-PLI measurements on non-human primate brains, because both the number and size of sections are significantly larger in humans. Therefore, it is a challenge to obtain measurements on a relatively larger number of sections and multiple brains. We decided to complete measurements on two brains from vervet monkeys to provide detailed fiber architecture of a non-human primate animal model first, while we continue to acquire data from human brain for future analyses. Furthermore, the 3D-PLI study on non-human primates has an advantage over human 3D-PLI study in the sense that it enables a direct comparison with the rich collection of tracer and dMRI measurements published over the last several decades (e.g. Schmahmann and Pandya, 2006; Schmahmann et al., 2007; Takemura et al., 2017).

While the reviewer notes that our choice of vervet monkey may limit the impact as compared with macaques, we would like to share our perspectives regarding the importance of studying vervet monkey as a non-human primate model. First, we would like to point out that while vervet monkeys are no more closely related to humans than are macaques, they are no more distantly related to humans than macaques. Therefore, we think that the diversity of non-human primate model species will help to assure that the elucidated fiber tracts are representative more broadly in old world monkeys, and not just a unique feature of a single species. Second, the vervet monkey is becoming increasingly important non-human primate model for neuroscience studies. The vervet monkey has been commonly examined in biomedical research as an earlier survey on 2004 (Carlsson et al., 2004). According to the survey by one of the authors (N.P.G.), there is an increasing number of scientific publications on the vervet monkey (Author response image 1). This increasing trend can be also seen the report of the National Institutes of Health’s report on non-human primate evaluation and analysis (2018; Figure 5; https://orip.nih.gov/sites/default/files/508%20NHP%20Evaluation%20and%20Analysis%20Final%20Report%20-%20Part%201.pdf). The recent growth in the use of vervets may be because of the fact that they pose fewer biosafety concerns, since they are not carriers of Herpes B virus (Baulu et al., 2002), are typically less expensive than macaques (Fremier et al., 2008) and emerging model of age-related disorders (Cramer et al., 2018; Latimer et al., 2019). Following this trend, there are increasing number of neuroscience studies investigating vervet monkey (Fears et al., 2009; 2011; Woods et al., 2011; Fedorov et al., 2011; Lundell et al., 2011; Dyrby et al., 2012; 2014; Maldjian et al., 2014; Menzel et al., 2019; Barrett et al., 2020). In sum, we would like to point out that (1) there is no disadvantage to study vervet monkey visual system regarding evolutionary distance from humans and (2) it is becoming much common to study the vervet monkey as a non-human primate model in neuroscience studies.

**Author response image 1. respfig1:** PubMed publications on scientific papers studying vervet monkeys. Vertical axis depicts a number of publications in each year (from 1940 to 2019). The survey was performed on PubMed (https://www.ncbi.nlm.nih.gov/pubmed/) by one of the author (N.P.G.).

Furthermore, we also note that we now provide evidence showing that the organization of the visual system is broadly similar between vervet and macaque monkeys. Author response image 2 depicts color-encoded principal diffusion direction map of ex vivo diffusion MRI data collected from vervet and macaque monkey brains. While there are small differences in the relative position of gyri and sulci, the topological arrangement of fiber bundles tested in this study (such as SS, VOF, and tapetum) are consistent between two species. We also note that detailed architecture of merging fibers, fiber terminations and short-range U-fibers are not visible in diffusion MRI data. Therefore, we think that there is a value to perform high-resolution 3D-PLI measurements on these bundles in the vervet monkey brain, and findings on vervet 3D-PLI data can be translated into the macaque because there is a rough inter-species correspondence of spatial organization of fiber tracts. We also note that, based on analysis of the receptor autoradiography data (Figure 12—figure supplement 1), parcellation of the occipital cortex is broadly similar between macaque and vervet brains. Therefore, since this evidence suggests that the visual system is largely similar between both species in terms of both fiber tracts and cortical areas, we believe that our choice of the vervet monkey will not limit the importance of this work.

**Author response image 2. respfig2:** Comparison of occipital white matter tracts visible in color-encoded diffusion MRI data between macaque and vervet monkeys. (**A**) Coronal view of color-encoded ex vivo diffusion MRI data collected from macaque monkey (left panel; voxel size, 0.25 mm isotropic; 121 directions; b = 4800 s/mm^2^ ; data is measured at National Institutes of Health and provided by D.A. Leopold and F.Q. Ye) and vervet monkey (right panel; 0.5 mm isotropic; 128 directions; b = 7700 s/mm^2^ ; data is measured at Danish Research Centre for Magnetic Resonance and provided by T.B. Dyrby following protocols described in Dyrby et al., 2011). The color scheme depicts the principal diffusion direction in each voxel (blue, superior–inferior; green, anterior–posterior; red, left–right). The details on these datasets have been described in previous publications (macaque data, Thomas et al., 2014; Reveley et al., 2015; Takemura et al., 2017; vervet data, Donahue et al., 2016). (**B**) Axial view of color-encoded ex vivo diffusion MRI data from macaque (left panel) and vervet monkey (right panel). Despite of inter-species difference in the position of gyri and sulci and differences in acquisition parameters across datasets, the position of stratum sagittale (SS), vertical occipital fascicle (VOF) and tapetum (T) were consistent between two species.

Specific changes: We have included a paragraph on the usefulness of the vervet monkey as a non-human primate model in the revised manuscript.

“Vervet monkey as a non-human primate model for neuroscience studies

In this study, we investigated the organization of fiber tracts in the visual system of vervet monkeys (*Chlorocebus aethiops sabaeus*). While historically the macaque monkeys (*Macaca mulatta*) have been widely tested in visual neuroscience studies, vervet monkeys became an increasingly important model for neuroscience studies because of its biosafety (Baulu et al., 2002), lower cost (Fremier et al., 2008) and similarity of age-related diseases with those of humans (Cramer et al., 2018; Latimer et al., 2019). In fact, there is an increasing number of neuroscience studies investigating vervet monkeys as a non-human primate model, including studies investigating fiber tracts (Fears et al., 2009; 2011; Woods et al., 2011; Fedorov et al., 2011; Lundell et al., 2011; Dyrby et al., 2013; 2014; Maldjian et al., 2014; Donahue et al., 2016; Menzel et al., 2019; Sarubbo et al., 2019; Barrett et al., 2020). We also note that while vervet monkeys are no more closely related to humans than are macaque monkeys, they are no more distantly related to humans than macaques. We think that the diversity of non-human primate model species will help to assure that the elucidated fiber tracts are representative more broadly of old world monkeys, not just a unique feature of a single species. Furthermore, our receptor autoradiography data also suggest cortical area of the visual system is broadly similar between vervet and macaque monkeys (Figure 12; Figure 12—figure supplement 1). Therefore, present investigation of vervet monkey visual system using 3D-PLI is an essential step toward the understanding of the organization of the primate visual system.”

Reviewer #3 raises a similar point, highlighting that the areas of the visual cortex are identified with reference to atlases, but these are in the macaque, not the vervet monkey. This should be acknowledged. It would be nice to see a bit more discussion on the autoradiography and criteria for defining areas, as this is highly specialized knowledge that we have to take at face value. At least a reference to the various criteria for identifying areas would be helpful.

We thank the reviewer for suggestions. In the revised manuscript, we further clarified our criteria for delineating the border between cortical areas, based on transmittance image of 3D-PLI data and receptor autoradiography. We also further clarified criteria derived from vervet and macaque data in each step. In the revised manuscript, we made an ample revision in the Methods concerning the identification of cortical areas, and also included a subsection describing the receptor autoradiography method. Finally, we note that we made 3D-PLI data in this work publicly available via the EBRAINS platform of the Human Brain Project (DOI: 10.25493/AFR3-KDK), in order to provide data unbiased by our border definition (see our response to comments on “Data availability” below).

Specific changes: In Materials and methods section, we wrote:

“Quantitative in vitro receptor autoradiography: experimental procedures and image acquisition and processing

Each of the frozen blocks was serially sectioned in the coronal plane (20 µm thickness) at -20°C using a cryostat microtome (CM 3050, Leica, Germany). Sections were thaw-mounted onto glass slides, and freeze-dried overnight. Adjacent sections were processed for histological staining of cell bodies (Merker, 1983) or myelin sheaths (Gallyas, 1979), or for quantitative in vitro receptor autoradiography following standardized protocols (Palomero-Gallagher and Zilles, 2018; Zilles and Palomero-Gallagher, 2017). The glutamatergic kainate receptor, cholinergic muscarinic M_2_ receptor and noradrenergic α_1_ receptor were visualized by incubating neighboring sections in solutions of respective tritiated receptor ligands. Non-specific binding was determined in parallel binding assays in which sections were incubated with the tritiated ligand and an unlabelled displacer. All binding assays were preceded by a 20-30 min preincubation in the respective buffer.

The kainate receptor was labelled with [^3^H]kainate (8 nM) in a 50 mM Tris-citrate buffer (pH 7.1) containing 10 mM Ca-acetate at 4°C for 45 min. The unlabeled displacer was kainate (100 μM). This main incubation was followed by a washing step with buffer (3 x 4 sec) and two dips into 100 ml aceton containing 2.5 ml glutaraldehyde.

The cholinergic muscarinic M_2_ receptor was labelled with [^3^H]oxotremorine-M (0.8 nM) in a 20 mM Hepes-Tris buffer (pH 7.5) containing 10 mM MgCl_2_ at 22°C for 60 min. The unlabeled displacer was carbachol (1 μM). The main incubation was terminated by a washing step with buffer (2 x 2 min) followed by a dip in distilled water at 4°C.

The noradrenergic α_1_ receptor was labelled with [^3^H]prazosin (0.2 nM) in a 50 mM Tris-HCl buffer (pH 7.4) at 30°C for 45 min. The unlabeled displacer was phentolamine (10 μM). The main incubation was terminated by a washing step with buffer (2 x 5 min) followed by a dip in distilled water at 4°C.

Following binding assays, sections were dried in a stream of warm air, then co-exposed together with standards of known concentrations of radioactivity to tritium-sensitive films (Hyperfilm, Amersham) for 4 (kainate receptor) or 6 (M_2_ and α_1_ receptors) weeks. After developing the films, autoradiographs were digitized with a CCD-camera (Axiocam MRm, Zeiss, Germany) and the image processing software Axiovision (Zeiss, Germany). A transformation curve indicating the relationship between grey values in the autoradiograph and concentrations of radioactivity in the tissue was computed for each receptor type using in-house-developed Matlab (The MathWorks, Natrick, MA) scripts and the standards with known radioactivity concentrations. Autoradiographs were then subjected to linear contrast enhancement, color coding and median filtering to provide a clear visualization of the regional and laminar receptor distribution patterns.”

We also wrote:

“Identification of cortical areas

The borders of visual areas V1, V2, V3V, V4V, V4T, V5/MT, and MST as well as of intraparietal areas LIPe, LIPi, VIP, and MIP can be identified in the vervet monkey based on differences in their myeloarchitecture as revealed in both transmittance images and fiber orientation maps (Zilles et al., 2016). These areas, as well as the areas defined in the parietal and temporal lobes are supported by comparisons between the brains of vervet monkey 1818, processed for PLI, and of vervet monkey 1695, processed for histological stainings and in vitro receptor autoradiography (Figure 12). Although fiber orientation maps and autoradiographs were obtained from different animals, the position of cortical areas identified by differences in myeloarchitecture as revealed by PLI (Figure 12A) is comparable to that of areas revealed by receptor distribution patterns (Figures 12B-D). Our parcellation of the vervet cortical ribbon is also supported by parallelisms between the regional and laminar distribution patterns of transmitter receptors in the vervet brain and those of the same receptors in sections obtained from comparable rostro-caudal levels in the macaque monkey brain (Figure 12—figure supplement 1). Furthermore, areas identified in the macaque brain have also been included in the atlases of Saleem and Logothetis, (2012) and Paxinos et al., (2009).

Given the correspondence of borders identified by analysis of PLI and receptor autoradiographic datasets in the vervet brain as well as the topological and receptor architectonical comparability of areas identified in the coronally sectioned vervet and macaque brains, cortical areas in the sagittally sectioned vervet monkey (ID 1947) were identified based on regional and laminar differences in the distribution of transmitter receptors in sagittal sections through the macaque brain (Zilles and Clarke, 1997) as well as by comparison with the atlases of Saleem and Logothetis, (2012) and Paxinos et al., (2009).

We have made 3D-PLI data used for generating figures publicly available via the EBRAINS platform of the Human Brain Project in order to provide data unbiased to our border definitions. The data (Axer et al., 2020) is accessible via the following DOI: 10.25493/AFR3-KDK.”

3) Summary of work.Reviewers #1 and #2 both point out that the authors do a fantastic job of characterizing these pathways from multiple perspectives. Yet the proverbial forest gets lost for the trees (i.e., it is hard to keep track of the details across images and text descriptions). Both reviewers feel that a visual summary of the trajectory of all the major pathways identified here would go a long way to assisting readers in understanding the take home message of the paper.

Following the reviewer’s suggestion, we included existing literature as well as a new figure highlighting previous dissection, tracer or diffusion MRI works to provide readers with an overview of approximate positions and trajectories of major tracts in the primate visual system (Figure 1 in the revised manuscript). We also included Table 1 in the revised manuscript with descriptions of the trajectory of major pathways (see our response to reviewer #2 below). We believe that the results of our present study improve our understanding on these tracts and Figure 1 highlights a need for a high-resolution approach as taken in this study in order to precisely understand the spatial organization of these tracts.

In Results section, we wrote.

“Figure 1 depicts descriptions of major tracts in the primate visual system, the stratum sagittale, the inferior longitudinal fascicle (ILF) and the vertical occipital fasciculus (VOF) as defined in previous human dissection studies (Figure 1A-B), and macaque tracer (Figure 1C) and dMRI (Figure 1D) studies. While these studies revealed the approximate position and trajectory of each tract (see Table 1 for definition of these tracts), none of them revealed the spatial organization of the whole single fiber tract in question at a micrometer resolution. Therefore, we aimed to investigate the detailed spatial organization of these tracts, and other tracts not well described in Figure 1, using 3D-PLI approach on the vervet monkey visual system.”

Reviewer #2 suggests something along the lines of the 3D representations from their 2016 paper, along with some black and white atlas-style coronal slices showing the boundaries of the different bundles. This reviewer pointed out that there are a number of potential readers who will not want to dive into the color schemes used in the current figures, but will want to know the placement of the bundles and general orientation of the fibers.

We understand that many readers are used to seeing the RGB color encoding scheme widely used in clinical diffusion MRI studies (e.g. Pajevic and Pierpaoli, 1999). However, we often find that this visualization scheme does not fully capture information of the 3D-PLI data well, since it assigns identical colors to fibers with orientations which are mirror-symmetric with respect to the vertical meridian. Therefore, we think that RGB color scheme often loses essential information in high-resolution 3D-PLI data. For this reason, an HSV color coding has been often used for PLI studies, even when the study aimed to compare PLI and dMRI (Henssen et al., 2019a; 2019b). We include Author response image 3 for a reference on the comparison between both color encoding schemes, and also include a citation for a previous work comparing color encoding schemes (Axer et al., 2011; Henssen et al., 2019a; 2019b) in the revised manuscript.

We also understand that it will be much preferred to obtain precise estimates of three-dimensional orientation distribution function and adapt them to visualizations. However, such estimation requires the extension of polarized microscopy with oblique measurements (Schmitz et al., 2018). The efficient protocol for such measurements is still under development. Please see our response to comments on the limitation of 3D-PLI below.

**Author response image 3. respfig3:** Comparison across color encoding schemes for 3D-PLI data. (**A**) HSV color representation of 3D-PLI data obtained from a medial sagittal section through the left hemisphere of the vervet monkey brain (ID1947; section #249). The image is identical to that presented in Figure 2A. (**B**) RGB color representation of the 3D-PLI data. The section is identical to that presented in panel A. The sphere indicates the RGB color coding of 3D fiber orientation in each pixel in panel B.

Specific changes: In Materials and methods section, we wrote:

“Comparison with other color encoding schemes often used in dMRI studies has been described in previous publications (Axer et al., 2011b; Henssen et al., 2019a; 2019b).”

In the Discussion section, we wrote:

“We summarize our observations in Table 1, while we note that all cortical terminations of each bundle may not be included in this table, since some were still difficult to identify due to crossing with superficial U-fibers or merging with other tracts.”

Reviewer #2 points out that an additional table summarizing entry/exit points for each of the various fiber bundles would be so useful. One of the advantages of this method over diffusion tractography is the improved ability to distinguish these-at 1.3μm resolution, 3D-PLI approaches the width of a single axon. It is fine that some of these are unclear because of crossing U-fibers, that can be included in the table.

We thank the reviewer for this suggestion. In the Discussion section we included Table 1, which is a summary of descriptions. Please note that as the reviewer mentioned, it is not possible to include all fiber tract cortical terminations.

Specific changes: Table 1 added.

In the Discussion section, we wrote:

“We summarize our observations in Table 1, while we note that all cortical terminations of each bundle may not be included in this table, since some were still difficult to identify due to crossing with superficial U-fibers or merging with other tracts.”

4) Clarity of findings.Reviewer #2 asks how confident are the authors that the VOF is actually primarily bringing fibers from dorsal to ventral brain areas, and not bringing dorsal and ventral fibers to the middle to traverse rostral/caudally (say, in the ILF?).

In both coronal and sagittal section data, we observed that VOF fibers connect dorsal and ventral extrastriate cortex, and we did not observe axons with sharp turns and terminating medially. Since false negatives are entirely possible in every experimental approach, we admit that we cannot discard the possibility that a small number of axons leaves the VOF and takes a rostral-caudal direction. In the revised manuscript, we included a discussion on this point.

Specific changes: In Discussion section, we wrote:

“While we cannot not rule out the possibility that a small number of axons may leave the VOF and take a rostro-caudal direction, our observation in 3D-PLI data supports the view that VOF fibers primarily travel along the superior-inferior axis and connect dorsal and ventral extrastriate cortex.”

Reviewer #3 points out that the authors might consider providing a bit more context in the Results section. Many of the tracts are discussed without any background and the logic of their decision to call a bundle by a certain name is left implicit. Surely, the authors have clear reasons to label their tracts, so it would be nice if some of this implicit-and no doubt enormous-reservoir of knowledge is shared. There is extensive review in the Discussion section, but the Results section could be made more readable. In general, more extensive reference to the literature would benefit the paper. For instance, there is repeated referenced to the 'controversy' around VOF, but this is supported by a citation to the paper that identified the tract-hardly a statement of controversy.

In response to reviewer’s comment and other comments, we made several changes to guide readers to understand the background of fiber tracts in the primate visual system. First, we included a figure in the Results section depicting previous literature of major tracts for guiding readers (Figure 1; see our response to reviewer #1 and #2’s comment, “Summary of work” above). Second, we include a table describing the name, location and trajectory, and cortical terminations observed in this study as Table 1 (see our response to reviewer #2’s comments on the table). Table 1 also summarizes the definition of each tract together with references to previous works which established the definition in question.

We also note that we have removed texts referring to “controversy”, following suggestions from reviewer #1 (see our responses to reviewer #1).

Finally, throughout the revision process, we noticed that the manuscript did not contain extensive references to previous works on the tapetum. We now included paragraphs in the Discussion section on the tapetum, with reference to previous works.

Specific changes: Figure 1 and Table 1.

In Discussion section, we wrote:

*“*Tapetum

While a number of neuroanatomists in the 19th century reported the existence of the tapetum, there has been substantial confusion regarding whether this fiber bundle should be considered as an association fiber or a callosal fiber (see Schmahmann and Pandya, 2006; 2007; Forkel et al., 2015 for historical debates on the tapetum). Among classical neuroanatomists, Burdach (1822) reported that the tapetum fibers are an extension of the splenium corpus callosi. Since this observation was later supported by a number of investigators using various methods, callosal origin of tapetum fibers may no longer be debated (Mettler, 1935; Clarke and Miklossy, 1990; see Schmahmann and Pandya, 2006 for a review). 3D-PLI data indeed directly visualized that tapetum fibers continue into splenium corpus callosI (Figure 11), confirming these previous works. 3D-PLI data further demonstrated the detailed course of tapetum fibers, namely its relative course with respect to neighboring fasciculi (SS and stratum calcarinum, as discussed below), lateral ventricle and ependyma (Figures 9C, 10C, 11 and Figure 9—figure supplement 1). This detailed information will provide essential insights for guiding dMRI-based tractography studies on splenium fibers, which have been considered to be relevant for important cortical functions such as reading (Binder and Mohr, 1992; Dougherty et al., 2007).”

5) Data availability.Reviewers #1 and 3, as well as the Senior Editor, feel that the data availability afforded by the authors does not meet the standards required by eLife. Reviewer #3 points out that the authors state that all data is presented in the figures, but of course that cannot even come close to demonstrating the richness of their data. In this era of open science we would very much hope they would create a resource for scientists based on their data; this would also be the only way that a reader can really evaluate their claims, as obviously the current manuscript by its nature is descriptive and reliant on the sections the authors choose to show. The reviewer appreciates that these data probably are so large as to make simple online sharing prohibitive, but the authors would be applauded for setting out a trajectory for sharing this type of data. At the very least they should rephrase their data availability statement.

We thank the editors and reviewers for this suggestion. As the reviewers imply, given the large file size of the 3D-PLI data, it is not practical for us to use widely-used public online sharing repositories. Building the infrastructure for sharing such large data is ongoing work in the Human Brain Project (HBP). In response to comments, we are making the 3D-PLI data analyzed in this manuscript publicly available via the EBRAINS platform of the Human Brain Project (DOI: 10.25493/AFR3-KDK). Furthermore, we included an e-mail address of one of corresponding authors in Research Centre Jülich (N.P.G.) to provide contacting information for scientists who may wish to obtain access to data not yet provided in the HBP platform.

Specific changes: In Data Availability statement, we wrote:

“Original data is publicly available via the EBRAINS platform of the Human Brain Project (Axer et al., 2020; DOI: 10.25493/AFR3-KDK).”

In Materials and methods section, we wrote:

“We have made 3D-PLI data used for generating figures publicly available via the EBRAINS platform of the Human Brain Project in order to provide data unbiased to our border definitions. The data (Axer et al., 2020) is accessible via the following DOI: 10.25493/AFR3-KDK.”